# Intron detention tightly regulates the stemness/differentiation switch in the adult neurogenic niche

Ainara González-Iglesias [1], Aida Arcas [1,9], Ana Domingo-Muelas [2,3,4,10,11], Estefania Mancini [5], Joan Galcerán [1,6], Juan Valcárcel [5,7,8], Isabel Fariñas [2,3] & M. Angela Nieto [1,6] ✉

The adult mammalian brain retains some capacity to replenish neurons and glia, holding promise for brain regeneration. Thus, understanding the mechanisms controlling adult neural stem cell (NSC) differentiation is crucial. Paradoxically, adult NSCs in the subependymal zone transcribe genes associated with both multipotency maintenance and neural differentiation, but the mechanism that prevents conflicts in fate decisions due to these opposing transcriptional programmes is unknown. Here we describe intron detention as such control mechanism. In NSCs, while multiple mRNAs from stemness genes are spliced and exported to the cytoplasm, transcripts from differentiation genes remain unspliced and detained in the nucleus, and the opposite is true under neural differentiation conditions. We also show that m⁶A methylation is the mechanism that releases intron detention and triggers nuclear export, enabling rapid and synchronized responses. m⁶A RNA methylation operates as an on/off switch for transcripts with antagonistic functions, tightly controlling the timing of NSCs commitment to differentiation.

The adult mammalian brain retains some capacity to replenish neurons and glia through life thanks to the lifelong persistence of neural stem cells (NSCs), the majority of which are found within the subependymal zone (SEZ) of the lateral ventricles. When NSCs become activated, they divide to produce either new NSCs or transit-amplifying progenitors (TAPs), which in turn give rise to neuroblasts[1]. Within the niche, the appropriate turnover of cells is achieved through the implementation of a plethora of regulatory mechanisms that ensure an accurate balance between the proliferation, self-renewal and differentiation of the NSC population. Recent studies analysing transcriptome dynamics during NSC activation and differentiation have evidenced that lineage progression is tightly controlled not only at transcriptional but also at post-transcriptional levels[2–4]. Indeed, NSCs transcribe both stemness and differentiation genes[4], pointing towards the existence of an additional but still unknown regulatory mechanism of post-transcriptional repression of the latter to ensure the maintenance of multipotency.

[1]Instituto de Neurociencias (CSIC-UMH), Sant Joan d'Alacant 03550, Spain. [2]Departamento de Biología Celular, Biología Funcional y Antropología Física and Instituto de Biotecnología y Biomedicina, Universidad de Valencia, Burjassot 46100, Spain. [3]Centro de Investigación Biomédica en Red sobre Enfermedades Neurodegenerativas (CIBERNED), 28029 Madrid, Spain. [4]Carlos Simon Foundation, 46980 Paterna, Valencia, Spain. [5]Centre for Genomic Regulation (CRG), The Barcelona Institute of Science and Technology, Barcelona 08003, Spain. [6]Centro de Investigación Biomédica en Red sobre Enfermedades Raras (CIBERER), 28029 Madrid, Spain. [7]Universitat Pompeu Fabra (UPF), 08003 Barcelona, Spain. [8]Institució Catalana de Recerca i Estudis Avançats (ICREA), 08010 Barcelona, Spain. [9]Present address: Department of Gene Therapy and Regulation of Gene Expression, Center for Applied Medical Research, University of Navarra, Pamplona 31008, Spain. [10]Present address: Department of Cell and Developmental Biology, Institute for Regenerative Medicine, Perelman School of Medicine, University of Pennsylvania, Philadelphia, PA 19104, USA. [11]Present address: Igenomix Foundation, 46980 Paterna, Valencia, Spain. ✉e-mail: anieto@umh.es

Although splicing and nuclear export were initially thought to be constitutive steps in gene expression, it is now clear that they can be highly selective, preventing or giving priority to the translation of particular transcripts in specific contexts (reviewed in[5,6]). Recent genome-wide studies have revealed the existence of transcripts that remain accumulated in the nucleus, due to the maintenance of unspliced introns in polyadenylated mRNAs[7–9]. Those introns, referred to as detained introns (DI) can be spliced in response to different signals[10–16], constituting an additional layer of post-transcriptional gene regulation.

Here we show that members of the Scratch family of transcription factors, known to promote neuronal differentiation[17–21] and to prevent cell death[18,22,23], are expressed in the adult subependymal neurogenic lineage and that Scratch1, in particular, is specifically required for neuronal differentiation. Interestingly, *Scratch1* is one of the differentiation genes transcribed in NSCs, but its transcripts are retained in the nucleus due to intron detention until differentiation is triggered. We find that in response to a neural differentiation signal, *Scratch1* mRNA is modified to contain N⁶-methyladenosine (m⁶A)[24] and then spliced and exported to the cytoplasm, where it can be translated. We also show that this mechanism regulates the subcellular localisation of other transcripts associated with NSC differentiation and interestingly, that a similar regulation of intron detention released by m⁶A modification occurs during differentiation in mRNAs transcribed from genes involved in the maintenance of stemness. This reveals a mechanism by which subsets of transcripts with critically opposing functions are alternatively retained in the nucleus. In summary, we describe intron detention as a novel mechanism to prevent the translation of differentiation genes in NSCs and that of stemness genes in differentiating neural cells, enabling fast and robust responses to either multipotency or differentiation signals.

## Results

### *Scratch1* mRNA accumulates in the nucleus of NSCs due to intron retention

We had previously shown that *Scratch1* is expressed in the wall of the lateral ventricles in the adult brain[25], where the SEZ is located. Using RNA-seq data obtained from the different cell populations isolated from the in vivo niche (GEO: GSE138243)[26] we found that *Scratch1* is already expressed in NSCs and that its expression gradually increases as cells differentiate into TAPs and progress into the neurogenic lineage (Fig. S1a). *Scratch1* expression in the neurogenic niche can be detected in glial fibrillary acidic protein (GFAP)-positive NSCs and in doublecortin (DCX)-expressing neuroblasts (Fig. 1a–c).

A careful analysis of the distribution of *Scratch1* transcripts, showed that they were remarkably condensed in NSCs (Fig. 1b), contrary to the situation in neuroblasts, where *Scratch1* mRNA was distributed throughout the cytoplasm (Fig. 1c). Double labelling with a nucleoporin antibody in cultured NSCs revealed that *Scratch1* mRNA was inside the nucleus (Fig. 1d, left panel; and S1b). In contrast, *Scratch2* mRNA exhibited the canonical cytoplasmic distribution (Fig. 1d, right panel). Moreover, in situ hybridisation for *Chromosomes 15*, where the *Scratch1* loci are located, showed a distribution compatible with the position of *Scratch1* mRNA foci in NSCs (Fig. S1c, compare with S1b), suggesting that the transcripts remain near its transcription sites when they accumulate in the nucleus. Altogether, these observations suggest that *Scratch1* mRNA export to the cytoplasm might be altered in NSCs.

As splicing is known to be a critical step in the export process[27], we wondered whether *Scratch1* nuclear transcripts had a defective splicing. Using intron-specific in situ hybridisation probes for *Scratch1* or *Scratch2* mRNAs, we detected the accumulation of *Scratch1* pre-mRNA in the nucleus (Fig. 1e, left panel), indicating that the transcripts remained unspliced in NSCs, thereby preventing their export to the cytoplasm and their translation. Conversely, *Scratch2* intron was only detectable in two small foci at the putative transcription sites

(Fig. 1e, right panel), indicating that *Scratch2* transcripts are co-transcriptionally spliced[28], consistent with its cytoplasmic distribution (Fig. 1d, right panel; and S1d, e). As the decrease in the splicing efficiency is often caused by inefficient recognition of the canonical splicing sites[29], we compared the sequence of the two paralogs focusing particularly on the splicing regions. We found an unusually long pyrimidine-rich (specially C-rich) region at the 3' region of the intron in *Scratch1* pre-mRNA, harbouring potential branch sites relatively distant to the 3' splice site AG, the more distal one preceded by U-rich stretches that may serve as the Polypyrimidine Tract (PPT) for the binding of U2AF2 (Fig. 1f). Thus, in contrast to *Scratch2* intron 1, which contains a more conventional 19 nt PPT, in *Scratch1* intron 1 the potential branch sites are separated from the 3' splice site by over 75/130 nucleotides (Fig. 1f). Moreover, *Scratch1* PPT is particularly rich in CC and CU dinucleotides, which also constitutes an unusual sequence feature at 3' splice sites that, together with its increased length, may be behind the retained intron phenotype.

Analysis of *Scratch* genes in different species showed that these unusual features are a common trait in mammals, including humans, whereas in other vertebrates PPT lengths were within the consensus range (Fig. S2a). Consistent with this, *Scratch1* paralogues in zebrafish (*scrt1a* and *scrt1b*), bearing a normal PPT length, exhibit a canonical cytoplasmic distribution in the palial germinal zone (PGZ; Fig. S2b), the area homologous to the neurogenic niches in rodents[30]. Interestingly, *scrt1a* and *scrt1b* are not transcribed in NSCs and their mRNAs are only detectable in the cytoplasm of HuC/D positive early neurons. Therefore, these results suggest that the progressive increase in the length of *Scratch1* PPT in mammals has generated an unusual 3' splice site region arrangement, allowing early transcription but deficient RNA processing and the inhibition of protein expression.

To assess whether *Scratch1* mRNA nuclear retention in NSC is directly caused by intron retention, we induced the overexpression of a spliced version of *Scratch1* mRNA in NSC primary cultures (Fig. 1g and S1f) and observed that, in addition to the transcripts retained in the nucleus that were also present in control cultures, *Scratch1* mRNA was detected in the cytoplasm of NSC (Fig. 1h, i). Altogether, these results indicate intron retention prevents the nuclear export of *Scratch1* mRNA in NSCs.

### Scratch1 favours the survival of the differentiating cells and their terminal differentiation into neurons

After characterising the expression pattern of *Scratch1* in the SEZ (Fig. 1a–c), we analysed its role during adult neurogenesis. We transduced NSCs primary cultures with lentiviruses containing *Scratch1*-specific shRNAs and artificially induced differentiation (Fig. 2a, b). On fully differentiated cultures (7DIV), we found that *Scratch1* downregulation led to a reduction in the production of neurons when compared to that in control cultures (Fig. 2c, middle panel), which was accompanied by an increase in the generation of astrocytes (Fig. 2c, top panel). This indicates that Scratch1 promotes neuronal differentiation, as previously shown in other systems[17–21], favouring neurogenesis at the expense of gliogenesis.

*Scratch1* downregulation also caused a significant increase in apoptosis (Fig. 2d, bottom panel; and 2e), which was already evident 2 days after the induction of differentiation (2DIV). Conversely, cell survival was not affected by *Scratch1* downregulation when cultures were kept in proliferation conditions (Fig. 2d, top panel; and 2e), indicating that the described role for Scratch in neuron survival in *C. elegans* and zebrafish[18,22,23] is also conserved in the mouse. However, protection from cell death appears to be dispensable when NSCs are maintained in an undifferentiated state. Moreover, as previously described for scratch2 in zebrafish[23], we found that Scratch1 promotes the survival of the differentiating NSCs acting downstream of p53, without affecting neither its levels (Fig. 2f, g) or *Mdm2* expression (Fig. 2h); but repressing the transcription of its target *Bbc3* (Fig. 2i), the main effector of p53-induced cell death in vertebrates[31].

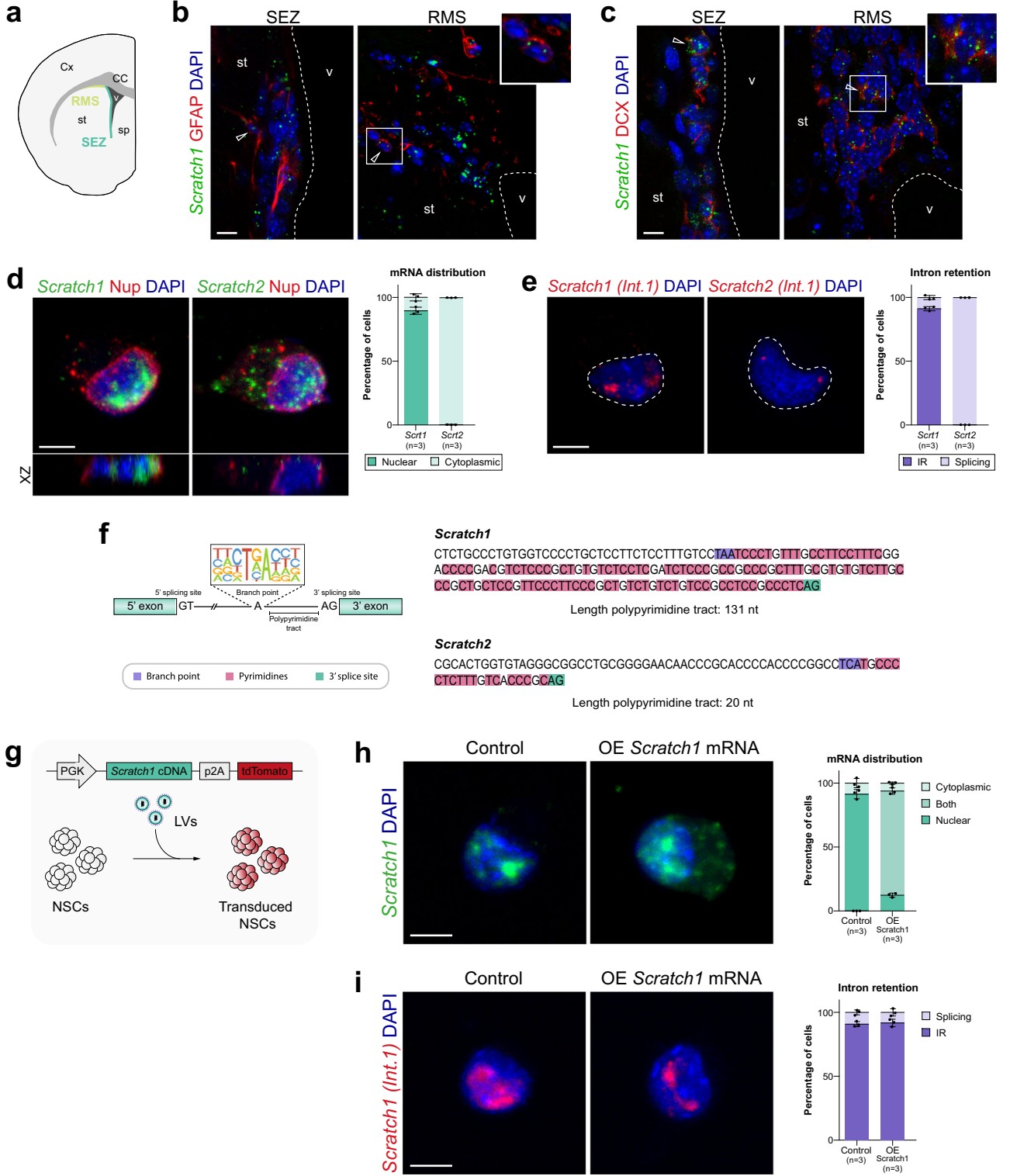

**Fig. 1 | *Scratch1* mRNA is retained in the nucleus of NSCs due to inefficient splicing. a** Schematic representation highlighting the location of the SEZ and RMS in a coronal section of an adult mouse brain hemisphere. In situ hybridisation for *Scratch1* (green) in the SEZ and RMS of adult mice (coronal sections), combined with immunohistochemistry for GFAP (NSCs, red, **b**) or DCX (neuroblasts, red, **c**); *n* = 9 mice over 3 independent experiments. **d** In situ hybridisation for *Scratch1* (left) and *Scratch2* (right) in NSCs in culture. The nuclear membrane is labelled using a pan-nucleoporin antibody (red). The window on the bottom of each panel shows the XZ orthogonal projection. Quantification of mRNA distribution in primary cultures (*n* = 3 mice). **e** In situ hybridisation for the only intron of *Scratch1* (left) and *Scratch2* (right) in NSCs in culture. Quantification of intron retention in primary cultures (*n* = 3 mice). **f** Comparison between the sequence of *Scratch1* and *Scratch2* polypyrimidine tracts. **g** Schematic drawing representing the transduction of NSCs with lentiviruses (LV) and the construct used for *Scratch1* overexpression experiments. **h** In situ hybridisation for *Scratch1* in control or *Scratch1* overexpressing NSC cultures. Quantification of mRNA distribution in primary cultures (*n* = 3 biologically independent samples). **i** In situ hybridisation for *Scratch1* intron in control or *Scratch1* overexpressing NSC cultures. Quantification of intron retention in primary cultures (*n* = 3 biologically independent samples). Scale bars: (**b**, **c**), 10 μm; and (**d**, **e**), (**h**, **i**), 5 μm. CC corpus callosum, Cx cortex, RMS rostral migratory stream, SEZ subependymal zone, sp septum, st striatum, v lateral ventricle. Data are presented as mean values ± SEM. Source data are provided as a Source Data file.

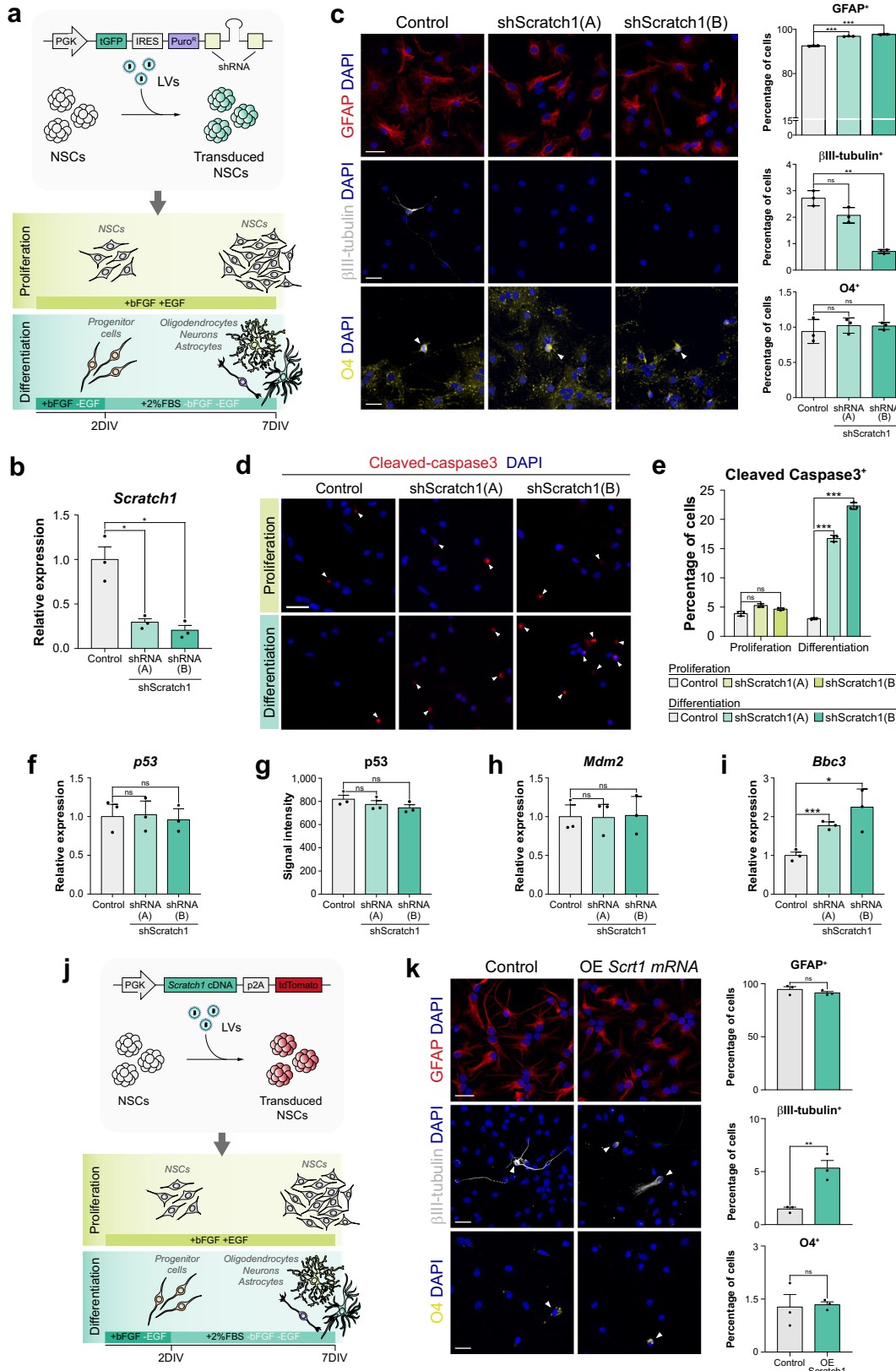

In agreement with our previous observations, we found that *Scratch1* overexpression in NSCs primary cultures induced an increase in the terminal differentiation of neurons (7DIV, Fig. 2j, k) and in the proportion of neuronal precursors present in the cultures during the process (2DIV, Fig. S3), both under differentiation and proliferation conditions, indicating that Scratch1 fosters NSC differentiation into neurons.

## *Scratch1* starts to be transcribed when stem cells acquire neural identity

The expression of *Scratch* genes is known to be neural specific[32], but considering that in NSCs, although *Scratch1* is transcribed, the protein cannot be translated, we wondered whether a similar situation occurred in pluripotent stem cells. We analysed its expression in

**Fig. 2 | Scratch1 promotes the survival of the differentiating cells and their terminal differentiation into neuros. a** Schematic representation of the transduction of NSCs with lentiviruses (LV) and the construct used for *Scratch1* loss of function experiments. Transduced NSCs were subsequently cultured either in proliferation or differentiation conditions. **b** Analysis of efficiency of the two independent *Scratch1* shRNAs by qPCR 2 days after the induction of NSC differentiation (2DIV; *p-value*(shA) = 0.018, *p-value*(shB) = 0.016, $n = 3$ biologically independent samples, by two-tailed Student's t-test). **c** Immunodetection and quantification of GFAP+ (red; *p-value*(shA) = 0.00006, *p-value*(shB) = 0.00005, $n = 3$ biologically independent samples, by two-tailed Student's t-test), βIII-tubulin+ (white; *p*-value(shA) = 0.19, *p*-value(shB) = 0.002, $n = 3$ biologically independent samples, by two-tailed Student's t-test) and O4+ (yellow; *p*-value(shA) = 0.70, *p*-value(shB) = 0.69, $n = 3$ biologically independent samples, by two-tailed Student's t-test) cells in differentiating cultures previously infected with control or shScratch1 lentiviruses 7 days after the induction of differentiation (7DIV). **d** Immunodetection of cleaved-caspase3 (red) in cultures of adult NSCs 2 days after plating the cells (2DIV), both in proliferation and differentiation conditions. **e** Quantification of the percentage of cleaved-caspase3+ cells in cultures infected with control or shScratch1 lentiviruses, both in proliferation and differentiation conditions (proliferation: *p*-value(shA) = 0.054, *p*-value(shB) = 0.182, differentiation: *p*-value(shA) = 0.00002, *p*-value(shB) = 0.00001, $n = 3$ biologically independent samples, by two-tailed Student's t-test). **f** Relative mRNA levels of *p53* in RNA extracts obtained 2 days after the induction of differentiation (2DIV; *p*-value(shA) = 0.50, *p*-value(shB) = 0.99, $n = 3$ biologically independent samples, by two-tailed Student's t-test). **g** Quantification of signal intensity for p53 immunofluorescence in NSCs fixed 2 days after the induction of differentiation (2DIV; *p*-value(shA) = 0.39, *p*-value(shB) = 0.16, $n = 3$ biologically independent samples, by two-tailed Student's t-test). Relative mRNA levels of (**h**) *Mdm2* (*p*-value(shA) = 0.85, *p* value(shB) = 0.46, $n = 3$ biologically independent samples, by two-tailed Student's t-test) and (**i**) *Bbc3* (*p*-value(shA) = 0.0008, *p*-value(shB) = 0.0244, $n = 3$ biologically independent samples, by two-tailed Student's t-test) relative to TBP in RNA extracts obtained 2 days after the induction of differentiation (2DIV). **j** Schematic drawing representing the transduction of NSCs with lentiviruses (LV) and the construct used for *Scratch1* overexpression experiments. Transduced NSCs were subsequently cultured either in proliferation or differentiation conditions. **k** Immunodetection and quantification of GFAP+ (red; *p*-value = 0.322, $n = 3$ biologically independent samples, by two-tailed Student's t-test), βIII-tubulin+ (white; *p*-value = 0.0065, $n = 3$ biologically independent samples, by two-tailed Student's t-test) and O4+ (yellow; *p*-value = 0.863, $n = 3$ biologically independent samples, by two-tailed Student's t-test) cells in control or *Scratch1* overexpressing NSC cultures 7 days after the induction of differentiation (7DIV). Arrowheads point to positive cells. Scale bars represent 25 μm. Data are presented as mean values ± SEM. ns not significant; *$p < 0.05$, **$p < 0.01$, ***$p < 0.001$. Source data are provided as a Source Data file.

embryonic stem cells (ESCs) and in induced pluripotent stem cells (iPSCs) reprogramed from NSCs isolated from the SEZ and observed that none of them expressed *Scratch1* (Fig. 3a, b).

To define the onset of *Scratch1* transcription, we induced the differentiation of ESC into NSC, using an ESC line that expressed GFP under the control of the *Sox1* promoter (*Sox1-GFP* knock-in 46C ESCs), allowing us to follow the progression of the differentiation process (Fig. 3c, d). We assessed this progression by the reduction in the expression of several pluripotency markers (*Nanog*, *Pou5f1* and *Zfp42*, Fig. 3e), and the concomitant increase in the expression of neural markers (*Sox1* and *Nestin*, Fig. 3f). In the case of *Scratch1*, its expression started to be detectable 5 days after the induction of differentiation, when the cells have already acquired the neural identity. Moreover, its level progressively increased until reaching its maximum at the end of the process (Fig. 3g). Therefore, these results confirmed that *Scratch* expression is exclusive of neural cells.

### The differentiation signal triggers *Scratch1* mRNA splicing and export to the cytoplasm

We have determined that *Scratch1* mRNA is not exported to the cytoplasm in NSCs, although we have also shown that the nuclear accumulation of the transcripts is transient, since in neuroblasts they present a cytoplasmic distribution (Fig. 1b, c). To better characterise the changes that occur in the distribution of the transcripts during NSC differentiation, neurosphere cells were seeded on Matrigel and deprived of EFG to trigger their differentiation (Fig. 4a)[33]. While in NSCs *Scratch1* mRNA was detected unspliced and localised inside the nucleus (Fig. 4b, top panel), upon the induction of differentiation the transcripts were processed and exhibited a canonical cytoplasmic distribution after 1 h (Fig. 4b, middle panel). This change in the splicing pattern and distribution of the mRNA was maintained for at least 7 days after the induction of differentiation (Fig. 4b, bottom panel; and S4a, b). Accordingly, we found that the proportion of spliced *Scratch1* mRNA significantly increased upon the induction of differentiation (Fig. 4d). As a control, we also analysed the distribution of *Scratch2* mRNA during the process and found that it was spliced at all the time points and, accordingly, localised in the cytoplasm (Fig. 4c, e); and the same distribution was observed in cultures were the spliced version of *Scratch1* mRNA was overexpressed (Fig. S4c–f).

Next, to confirm that the changes observed in the processing and distribution of *Scratch1* mRNA were a response to EGF withdrawal, we treated the cells with two different EGF receptor signalling inhibitors (Gefitinib and Afatinib; Fig. 4f). As when NSCs were cultured in the absence of EGF, the treatment with these inhibitors resulted in the splicing of the mRNA (Fig. 4g), the loss of *Scratch1* transcripts nuclear accumulation and their export to the cytoplasm (Fig. 4h). Therefore, these results indicate that *Scratch1* transcripts are rapidly spliced and exported in response to the differentiation signal, and also reveal that the retention of the intron in *Scratch1* mRNA is reversible (Fig. 4b, d). Thus, this can be considered an event of intron detention in NSCs which is the result of regulated splicing rather than simply slow processing by reduced efficiency in the recognition of the splicing site[8].

### N⁶-methyladenosine modification controls the processing and export of *Scratch1* mRNA in response to the differentiation signal

Having determined that *Scratch1* mRNA splicing is detained until NSCs receive the differentiation signal, we then wanted to study the mechanism that controls this switch in the processing and subcellular localisation of the transcripts. We focused on post-transcriptional mRNA modifications, which have been involved in the regulation of almost every post- transcriptional step in gene expression[34], and we examined N6-methyladenosine (m⁶A), which is the most prevalent epitranscriptomic modification and it is known to regulate both splicing[35–38] and nuclear export[39–42].

As a first approach we performed an in silico analysis of *Scratch1* pre-mRNA sequence and found the GGm⁶ACU consensus motif (Fig. 5a)[43] in numerous positions, some of which exhibited high probability of being methylated considering the features of the surrounding sequence, especially located in the intronic region and in the 3'UTR (Fig. 5b).

m⁶A is a dynamic modification that can be potentially added and removed in the lifetime of a single mRNA molecule, and, hence, the methylation of a given m⁶A site is highly dependent on the cellular context[43]. To analyse whether *Scratch1* mRNA is methylated during adult neurogenesis and whether methylation changes with NSC differentiation, we performed an m⁶A-RNA immunoprecipitation assay (m⁶A-RIP; Fig. 5c) and observed that there is an enrichment in *Scratch1* mRNA upon m⁶A-RIP compared to the input, indicating that these transcripts were methylated (Fig. 5d). We used *Actb* mRNA as a positive control since it is known to be constitutively methylated[44]. In addition, we observed an increase in *Scratch1* mRNA methylation upon the

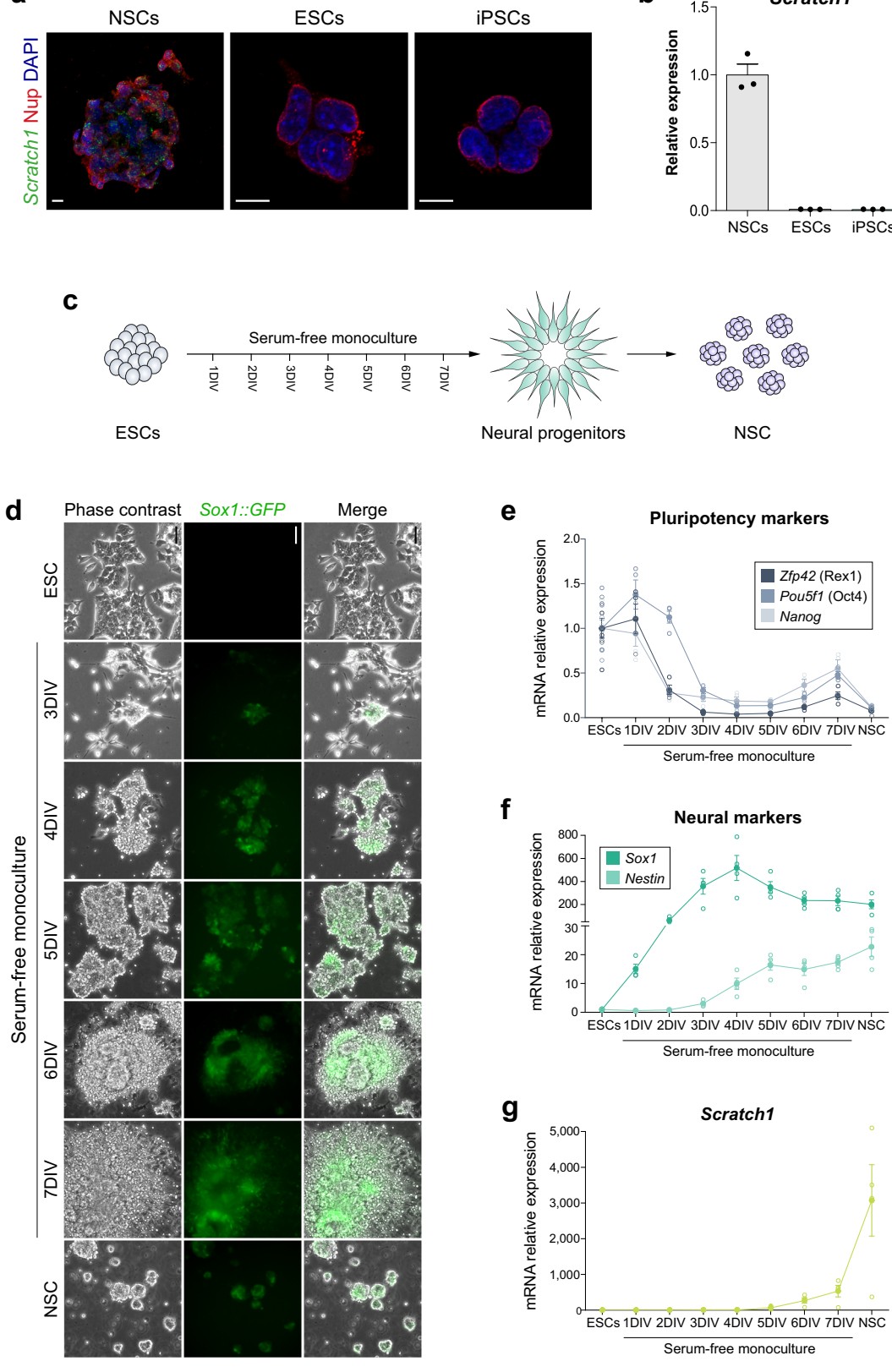

**Fig. 3 | *Scratch1* expression is restricted to the neural lineage. a** In situ hybridisation for *Scratch1* (green) combined with immunohistochemistry for nucleoporins (red) in cultured NSCs (left), ESCs (center) and iPSCs (right). **b** Relative mRNA levels of *Scratch1* in RNA extracts obtained from NSCs, ESCs and iPSCs (*n* = 3 biologically independent samples). **c** Schematic representation of the protocol to obtain ESC-derived NSCs. **d** Cells at different time points of the ESC-to-NSC differentiation process observed under phase contrast optics. GFP (green) labels the cells that express Sox1. Relative mRNA levels of several pluripotency markers (*Nanog, PouSf1* and *Zfp42*, **e**); neural markers (*Sox1* and *Nestin*, **f**); and *Scratch1* (**g**) in RNA extracts obtained at different time points during the differentiation of ESCs into NSCs (*n* = 4 independent experiments). Scale bars: (**a**),10 μm; and (**d**), 100 μm. DIV, days in vitro. Data are presented as mean values ± SEM. Source data are provided as a Source Data file.

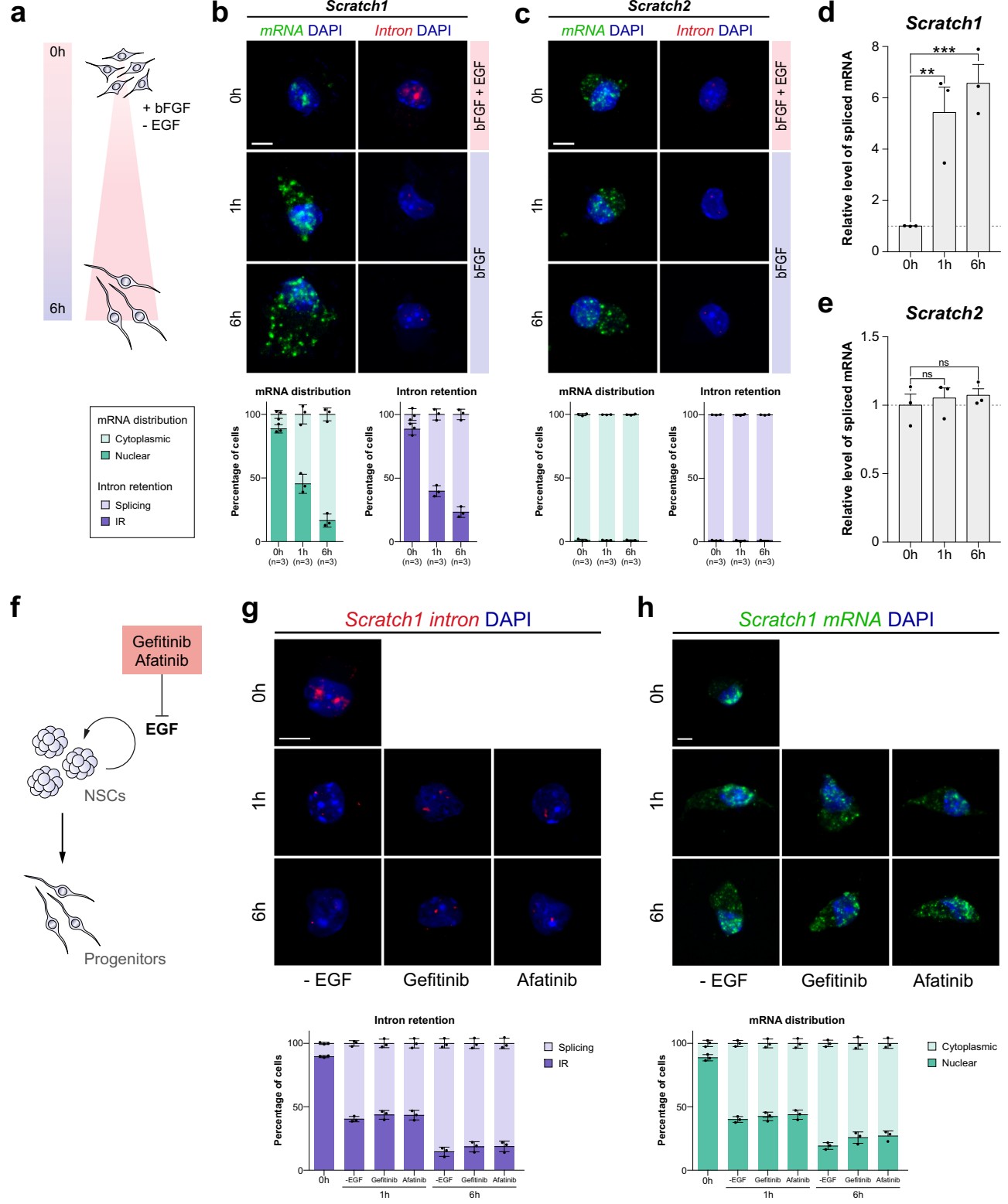

induction of NSC differentiation, coinciding with the processing and the export of the transcripts (Fig. 5d, e), suggesting that RNA methylation can regulate these two processes in the context of adult neurogenesis. This was confirmed after treating NSCs with 3-deazaadenosine (DAA), a strong inhibitor of m⁶A (Fig. 5f)[45]. In the presence of DAA, *Scratch1* mRNA failed to be spliced and exported in response to the induction of NSC differentiation, whereas in control cultures the transcripts were processed and exhibited a cytoplasmic localisation (Fig. 5g–i). The same was observed when m⁶A deposition was inhibited by *Mettl3* downregulation (Fig. S5). Moreover, as expected, when *Scratch1* cDNA was overexpressed in NSC and cultures were treated with DAA, we observed that some transcripts escaped nuclear retention and distributed throughout the cytoplasm (Fig. 5j–l). Therefore, altogether these observations link RNA methylation with *Scratch1* mRNA splicing and export to the cytoplasm during NSC differentiation.

**Fig. 4 | *Scratch1* mRNA is spliced and exported in NSCs in response to the differentiation signal. a** Schematic representation of the protocol used to induce the differentiation of NSCs in culture. **b** In situ hybridisation for *Scratch1* mRNA (green) and *Scratch1* intron (red) in NSCs at 0 h, 1 h and 6 h after the induction of differentiation. Quantification of mRNA distribution and intron retention (*n* = 3 mice). **c** In situ hybridisation for *Scratch2* mRNA (green) and *Scratch2* intron (red) in NSCs at 0 h, 1 h and 6 h after the induction of differentiation. Quantification of mRNA distribution and intron retention (*n* = 3 mice). **d** Ratio of spliced *Scratch1* mRNA in NSCs at 0 h, 1 h and 6 h after the induction of differentiation (*p*-value(1 h) = 0.010, *p*-value(6 h)=0.001, *n* = 3 mice, by two-tailed Student's t-test). **e** Ratio of spliced *Scratch2* mRNA in NSCs at 0 h, 1 h and 6 h after the induction of differentiation (*p*-value(1 h)=0.824, *p*-value(6 h) = 0.370, *n* = 3 mice, by two-tailed Student's t-test). **f** Schematic representation of the alternative protocol used to induce neural differentiation, the treatment of NSCs with EGF inhibitors. **g** In situ hybridisation for *Scratch1* intron (red) in NSCs at 0 h, 1 h and 6 h after the induction of differentiation (left) or treatment with Gefitinib (center) or Afatinib (right). Quantification of intron retention (*n* = 3 mice). **h** In situ hybridisation for *Scratch1* mRNA (green) in NSCs at 0 h, 1 h and 6 h after the induction of differentiation (left) or treatment with Gefitinib (center) or Afatinib (right). Quantification of mRNA distribution (*n* = 3 mice). Scale bars represent 5 μm. Data are presented as mean values ± SEM. ns not significant; *p < 0.05, **p < 0.01, ***p < 0.001. Source data are provided as a Source Data file.

## Intron detention is a general mechanism that regulates the translation of multiple transcripts associated with NSC self-renewal and differentiation

We have found that intron detention, linked to mRNA methylation, regulates the subcellular localisation of *Scratch1* transcripts, controlling their availability for translation.

Next, to assess whether other transcripts were also post-transcriptionally regulated by the same mechanism, we performed bulk RNA-seq at different time points during NSC differentiation and analysed the data using Vertebrate Alternative Splicing and Transcription Tools (VAST-TOOLS), a toolset for profiling and comparing alternative splicing events in RNA-seq data (Fig. 6a), and with it, events of intron retention (IR).

We first analysed changes in gene expression at the transcriptional level and detected the up-regulation of genes associated with neural differentiation together with the downregulation of NSC-related genes, indicating that differentiation was efficiently induced (Fig. 6b). On the other hand, when we performed a hierarchical clustering analysis based on the differentially expressed genes (DEGs), we observed that although the samples obtained 6 h after the induction of NSC differentiation clustered together, the samples obtained from undifferentiated NSCs (0 h) and cells after 1 h of differentiation did not cluster according to their time-point (Fig. 6c), indicating that 1 h was not sufficient to generate significant changes at the transcriptional level in this system. However, when hierarchical clustering was performed based on the IR events (Extended data 1) the samples clustered according to the time after induction of differentiation (Fig. 6d). Thus, as expected, if splicing regulates differentiation, changes in splicing patterns precede changes in transcriptional patterns and better characterise, in molecular terms, the phenotypic status of the different differentiation stages.

To identify the transcripts regulated by intron detention in the context of adult neurogenesis, we applied soft clustering based on the fuzzy c-means algorithm, which revealed several patterns of splicing (Fig. S6a). We focused on Cluster 1, which included genes that presented the same splicing pattern as *Scratch1* mRNA (i.e. intron detention in NSCs and splicing in response to the differentiation signal; Fig. 7a, b) and on Cluster 2, which was composed by genes with the opposite pattern (i.e. normal processing of the transcripts in NSCs and intron detention upon the induction of differentiation; Fig. 7c, d). Functional enrichment analysis showed that Cluster 1 contained genes associated with both neuronal and glial differentiation, as well as with axonogenesis and axon guidance (Fig. 7a); whereas genes in Cluster 2 were associated with stem cell maintenance and regulation of proliferation (Fig. 7c). As previously reported for detained introns in other systems[7,8], we observed that Cluster 1 and Cluster 2 introns were on average shorter (Fig. S6b) and exhibited a higher G/C content (Fig. S6c) than constitutively spliced introns.

Additionally, we performed the same analyses using publicly available RNA-seq datasets obtained from NSCs, early neuroblasts (ENB) and late neuroblasts (LNB) FACS-isolated from the SEZ or OB of adult mice (GEO: GSE944991)[4]. We found several splicing patterns (Fig. S7a), among which we identify two clusters that, combined, were equivalent to Cluster 2 described here (Cluster 1 + 4, Fig. S7b and Fig. 7c); and one cluster that recapitulated the trend of Cluster 1 (Cluster 6, Fig. S7c and Fig.7a). Interestingly, we identified a clear shared functional enrichment despite the reduced overlap between the individual IR-regulated genes in each dataset. Cluster 1 + 4, as Cluster 2, contained genes involved in stem cell maintenance and regulation of proliferation (Fig. S7b, c). By contrast, genes in Cluster 6 were associated with neuronal and glial differentiation, axonogenesis and axon guidance, as we have found for Cluster 1 (Fig. S7c, a). Therefore, these observations indicate that the intron detention regulatory mechanism that we describe here also operates in the SEZ for the regulation of adult neurogenesis in vivo.

To validate the detected events of intron detention and their impact on subcellular localisation of the transcripts, we selected representatives of each cluster and examined the distribution of their mRNAs in response to NSC differentiation conditions in culture. On the one hand, transcripts from Cluster 1 were unspliced and consistently accumulated in the nucleus in NSCs and they were processed and exported upon induction of differentiation (Fig. 7e–i and S8), as indicated by the RNA-seq data (Fig. 7f). Moreover, when cells were treated with DAA, transcripts did not undergo splicing and remained in the nucleus, indicating that, as observed for *Scratch1* transcripts, mRNA methylation promoted the splicing and export of the transcripts included in Cluster 1. On the other hand, for transcripts representative of Cluster 2, we observed that their mRNAs were normally spliced in NSCs, while intron detention was detected when the differentiation of these cells was induced (Fig. 7g, j and S9), as the RNA-seq data indicated (Fig. 7h). Consequently, the transcripts of these genes exhibited a canonical cytoplasmic distribution in NSC and their accumulation in the nucleus was observed during differentiation. Furthermore, although the treatment with DAA had no additional effect on the processing and export of the transcripts of this cluster, which were already unspliced and accumulated in the nucleus in response to the differentiation signal, RNA methylation inhibitors prevented the splicing and export of these mRNAs in undifferentiated NSCs, indicating that the subcellular localisation of the transcripts that present intron detention upon the induction of differentiation (Cluster 2) is also regulated by mRNA methylation. The same effect was observed when m⁶A deposition was inhibited by *Mettl3* downregulation instead of by DAA treatment (Fig. S10).

In addition, according to the observations in cultured cells, Cluster 1 transcripts were found accumulated in the nucleus of NSCs in the SEZ, whereas they presented a cytoplasmic distribution in neuroblasts (Fig. S11a–c). By contrast, Cluster 2 transcripts were observed in the cytoplasm of NSC from the SEZ, while they accumulated in the nucleus in neuroblasts (Fig. S11d–f). Therefore, these results indicate that mRNA methylation promotes the processing and export of transcripts of both stemness and differentiation genes, and that intron detention prevents the availability for translation of numerous transcripts relevant for differentiation in NSCs and for self-renewal in differentiating cells in the context of adult neurogenesis.

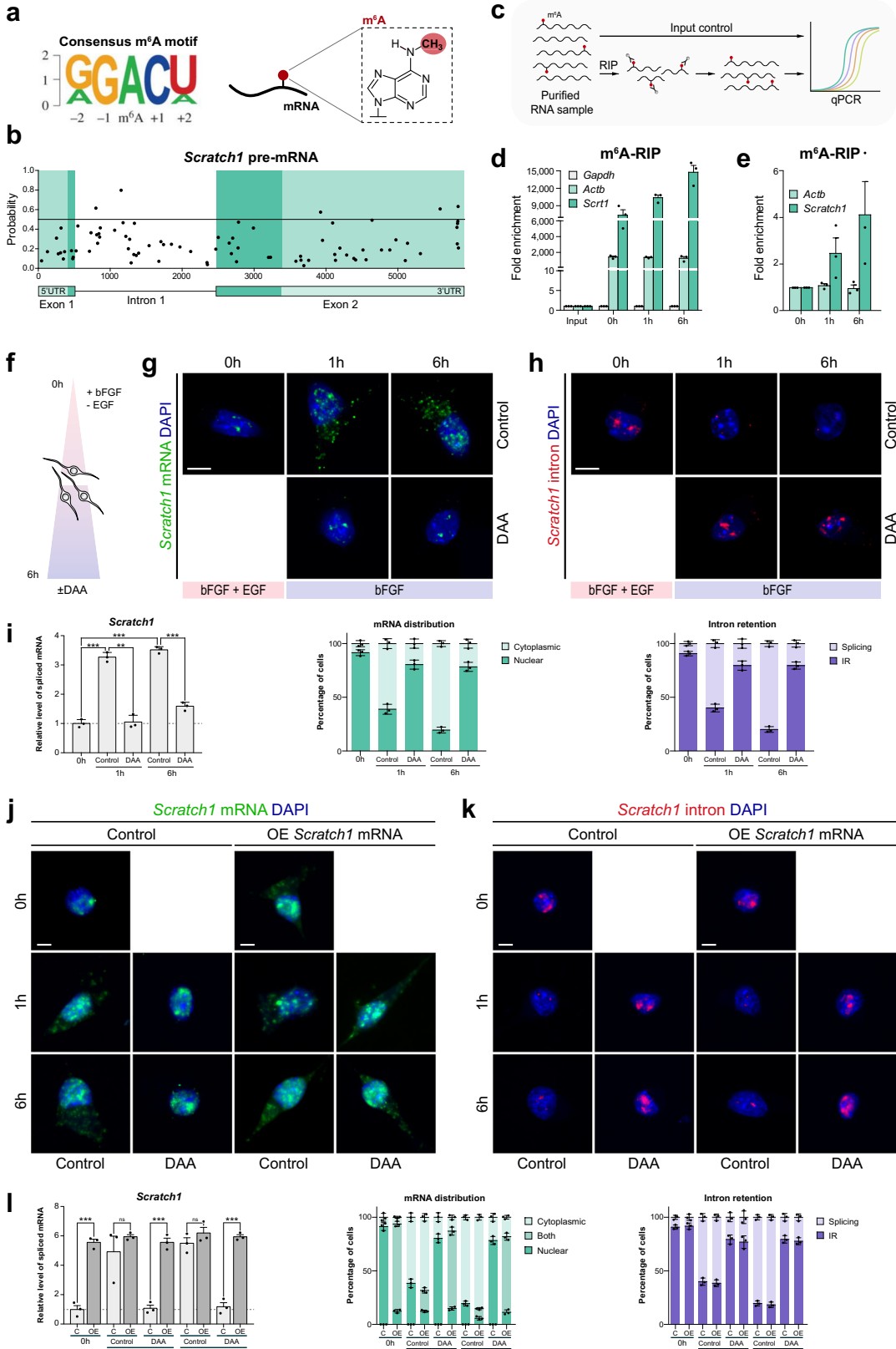

## m⁶A epitranscriptomic modification regulates the balance between NSC self-renewal and differentiation

To better understand the role of RNA methylation on the regulation of NSC behaviour, we transduced primary cultures of NSCs with lentiviruses containing a non-silencing control or a *Mettl3*-specific shRNA

(Fig. 8a). Mettl3 downregulation caused a significant increase in cell death, both under proliferation or differentiation conditions (Fig. S12), consistent with the implication of m⁶A in the regulation of cell homoeostasis and survival[46]. Regarding neurogenesis, we found that *Mettl3* downregulation significantly increases the generation of

**Fig. 5 | RNA methylation promotes *Scratch1* mRNA splicing and export during neural differentiation. a** Sequence logo representing the consensus motif for m⁶A: RRm⁶ACH (R = A or G, G > A; H = A, U or C, U > A > C). **b**, In silico analysis of methylation probability in the different putative m⁶A sites present in *Scratch1* pre-mRNA. **c** Schematic representation of the m⁶A-RIP protocol. **d** m⁶A-RIP-qPCR for *Gapdh*, *Actb* and *Scratch1* at 0 h, 1 h and 6 h after the induction of differentiation (*n* = 3 biologically independent samples). **e** m⁶A-RIP-qPCR for *Actb* and *Scratch1* at 0 h, 1 h and 6 h after the induction of differentiation: enrichment in *Scratch1* levels compared to *Actb* (*n* = 3 biologically independent samples). **f** Schematic representation of the protocol used to induce the differentiation of NSCs in culture. **g** In situ hybridisation for *Scratch1* mRNA (green) in control NSCs and in NSCs treated with 3-deazaadenosine (DAA). Quantification of mRNA distribution (*n* = 3 biologically independent samples). **h** In situ hybridisation for *Scratch1* intron (red) in control NSCs and in NSCs treated with 3-deazaadenosine (DAA). Quantification of intron retention (*n* = 3 biologically independent samples). **i** Ratio of spliced *Scratch1* mRNA in control and DAA-treated NSCs at 0 h, 1 h and 6 h after the

induction of differentiation (control: *p*-value(1 h) = 0.0004, *p*-value(6 h) = 0.0001, DAA: *p*-value(1 h) = 0.0013, *p*-value(6 h)=0.0003, *n* = 3 biologically independent samples, by two-tailed Student's t-test). **j** In situ hybridisation for *Scratch1* mRNA (green) in control or *Scratch1* overexpressing NSC cultures (±DAA) at 0 h, 1 h and 6 h after the induction of differentiation. Quantification of mRNA distribution (*n* = 3 biologically independent samples). **k** In situ hybridisation for *Scratch1* intron (red) in control or *Scratch1* overexpressing NSC cultures (±DAA) at 0 h, 1 h and 6 h after the induction of differentiation. Quantification of intron retention (*n* = 3 biologically independent samples). **l** Ratio of spliced *Scratch1* mRNA in control or *Scratch1* overexpressing NSC cultures (±DAA) at 0 h, 1 h and 6 h after the induction of differentiation (*p*-value(0 h) = 0.0001, *p*-value(1 h,C) = 0.3968, *p*-value(1 h,DAA) = 0.0002, *p*-value(6 h,C) = 0.2620, *p*-value(6 h,DAA) = 0.0001, *n* = 3 biologically independent samples, by two-tailed Student's t-test). Scale bars represent 5 μm. Data are presented as mean values ± SEM. ns not significant; *\**p* < 0.05, *\*\**p* < 0.01, *\*\*\**p* < 0.001. Source data are provided as a Source Data file.

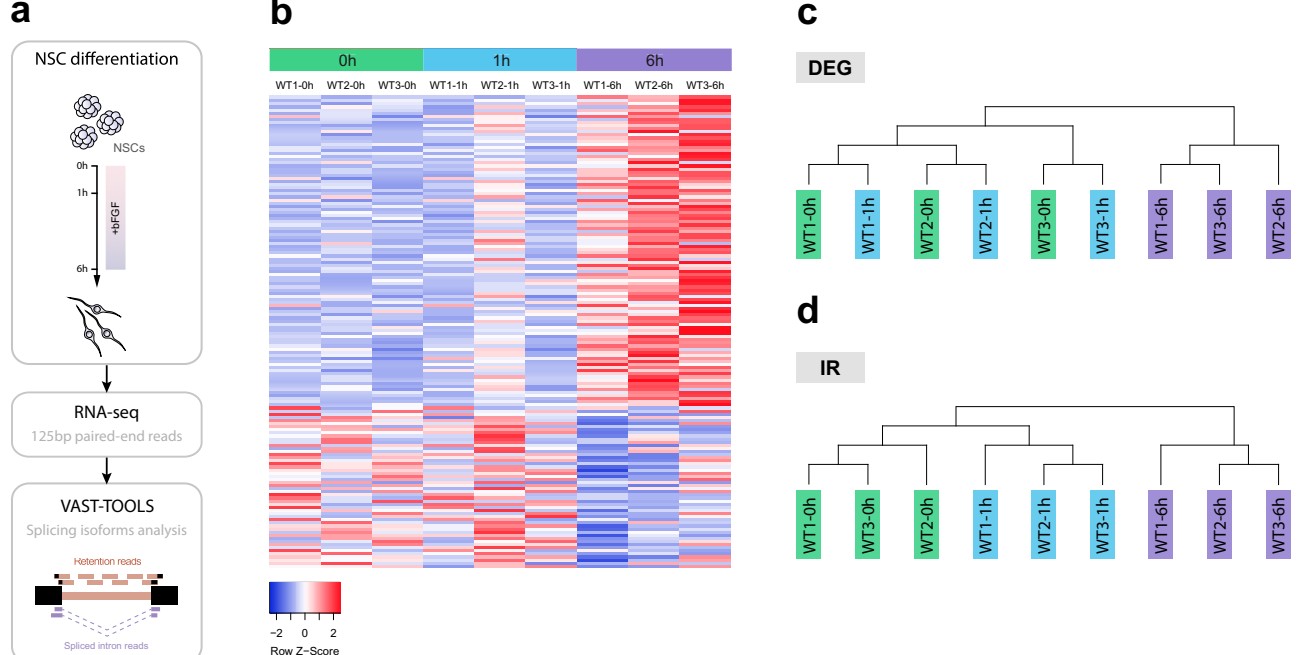

**Fig. 6 | Intron retention and the stemness-differentiation switch. a** Strategy used to identify additional genes regulated by intron detention. **b** Heatmap showing changes in the expression of genes known to change during neural differentiation at 0 h, 1 h and 6 h after induction. **c** Samples grouped using hierarchical clustering based on differentially expressed genes (DEG). **d** Samples grouped using hierarchical clustering based on intron retention events (IR).

neuroblasts and neurons when cultures were maintained in proliferation conditions (Fig. 8b, c, top panels), indicating that m⁶A favours the maintenance of NSCs in an undifferentiated state. However, upon the induction of differentiation, we observed that the proportion of both DCX⁺ and β-tubulin⁺ cells was reduced compared to the control (Fig. 8b, c, bottom panels), indicating that m⁶A is also required for neurogenesis and suggesting that RNA methylation contributes to the balance between NSC self-renewal and differentiation. Accordingly, as a consequence of *Mettl3* downregulation, we observed a reduction in the generation of astrocytes (GFAP⁺; Fig. 8d, top panel), neurons (β-tubulin⁺; Fig. 8d, middle panel) and oligodendrocytes (O4⁺; Fig. 8d, bottom panel) in terminally differentiated cultures. This confirms that m⁶A deposition is essential for the proper differentiation of NSCs.

Altogether, these results indicate that mRNA methylation, through the regulation of splicing and nuclear export of multiple mRNAs belonging to antagonistic functional groups, controls the availability for translation of transcripts relevant for NSC self-renewal and differentiation, contributing to the fine-tuning of adult neurogenesis (Fig. 8e).

## Discussion

Several transcriptomic analyses performed during the progression of NSCs into the neurogenic lineage have revealed that mRNAs typically associated with more differentiated cells can be already detected in NSCs[2,3,47]. Moreover, it has been recently shown that in this context some transcripts are translated less efficiently than expected considering their mRNA levels[4]. Altogether, these observations point to an important influence of an unknown post-transcriptional regulation in the control of adult neurogenesis. Here we describe a novel regulatory mechanism that controls the subcellular localisation of a large number of transcripts associated with both the maintenance of NSC in an undifferentiated state and with their differentiation. This mechanism can tightly regulate the availability of these mRNAs for translation, contributing to the precise temporal control of neural differentiation and avoiding potential conflicts in fate decisions due to non-specific transcription.

Splicing has been identified as a relevant process for adult neurogenesis[48]. In this study, we have detected a subset of transcripts that present intron detention in NSCs. In the case of the only intron of

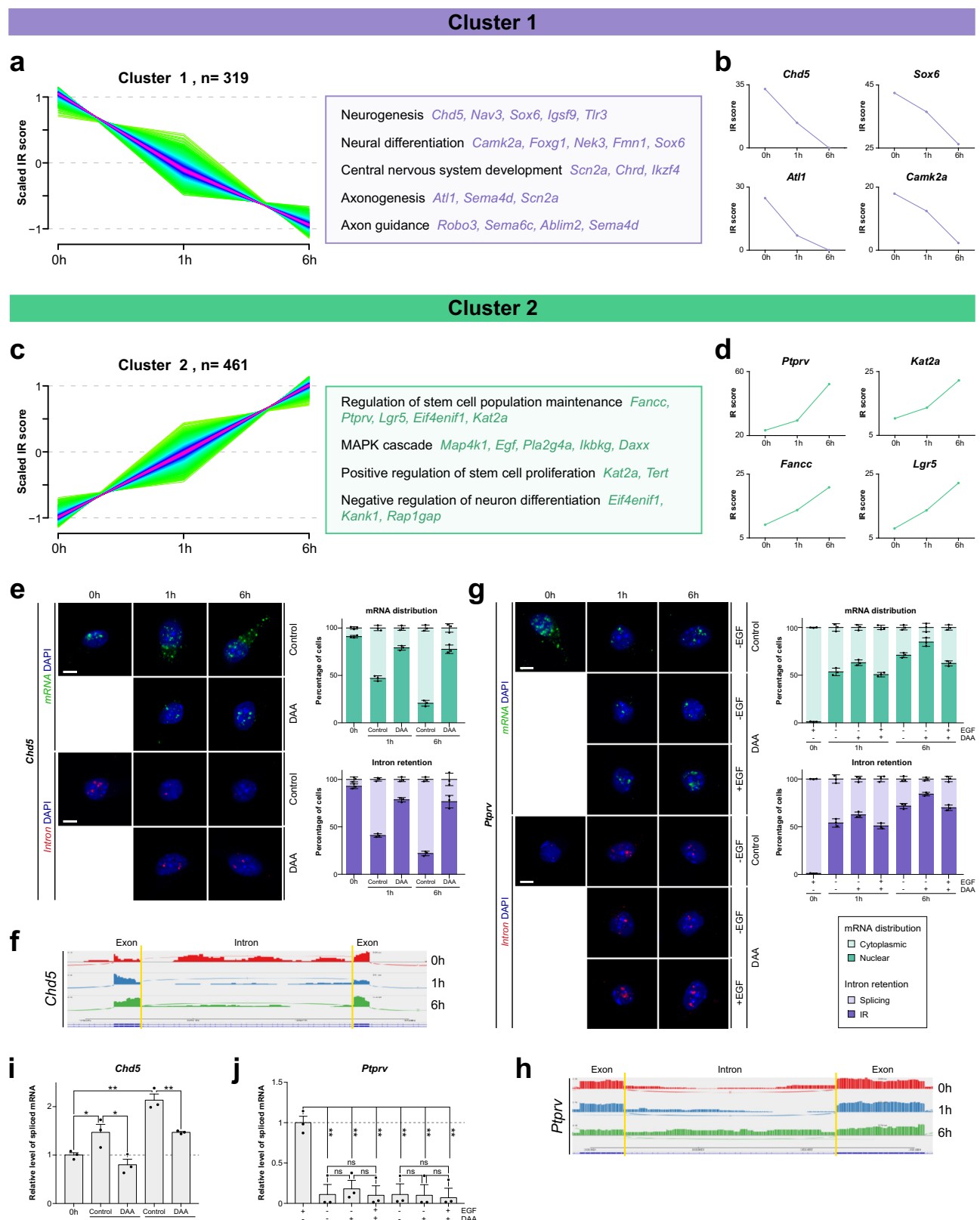

*Scratch1*, we have observed that one of the regions that must be recognised by the spliceosome, the PPT, has undergone a progressive enlargement during evolution, likely linked to its mechanism of regulation. The unusual sequence features of this region, including its C-richness (e.g. high proportion of CC and UC dinucleotides), relative frequency of UUCU/C (PTB repressive motifs at PPTs)[49] and potential

branch sites at distant positions from the 3' splice site AG, may be part of the regulatory signals that contribute to intron detention[29,50], allowing the implementation of an additional layer of expression regulation. In agreement with this, we have observed that in mammals, *Scratch1* starts to be transcribed when stem cells acquire neural identity, although the mRNA remains in the nucleus, unavailable for

**Fig. 7 | Intron detention regulates the subcellular localisation of multiple transcripts to control the balance between stemness and differentiation during adult neurogenesis. a** Plot illustrating time point-specific changes in intron detention for genes belonging to Cluster 1 (left) and representative Gene Ontology (GO) terms of the biological process categories enriched in this cluster (right). **b** Intron retention score for genes representative of Cluster 1: *Chd5*, *Sox6*, *Atl1* and *Camk2a*. **c** Plot illustrating time point-specific changes in intron detention for genes belonging to Cluster 2 (left) and representative Gene Ontology (GO) terms of the biological process categories enriched in this cluster (right). **d** Intron retention score for genes representative of Cluster 2: *Kat2a*, *Lgr5*, *Fancc* and *Ptprv*. **e** In situ hybridisation for *Chd5* mRNA (green) or intron (red) in control or NSCs treated with DAA, taken as an example of a Cluster 1 gene. Quantification of mRNA distribution and intron retention ($n = 3$ biologically independent samples). **f** Sashimi plots depicting read density and number of splice junctions for *Chd5* (Cluster 1) at different time points of the differentiation process, showing differential intron detention, which decreases with differentiation. **g** In situ hybridisation for *Ptprv*

mRNA (green) or intron (red) in control NSCs and in NSCs treated with DAA, as an example of a gene from Cluster 2. Quantification of mRNA distribution and intron retention ($n = 3$ biologically independent samples). **h** Sashimi plots depicting read density and number of splice junctions for *Ptprv* (Cluster 2) at different time points of the differentiation process, showing an increase in intron detention with differentiation. **i** Ratio of spliced *Chd5* mRNA in control and DAA-treated NSCs at 0 h, 1 h and 6 h after the induction of differentiation (control: $p$-value(1 h) = 0.05, $p$-value(6 h) = 0.0012, DAA: $p$-value(1 h) = 0.036, $p$-value(6 h) = 0.008, $n = 3$ biologically independent samples, by two-tailed Student's t-test). **j** Ratio of spliced *Ptprv* mRNA in control and DAA-treated NSCs at 0 h, 1 h and 6 h after the induction of differentiation (-EGF: $p$-value(1 h, C) = 0.0029, $p$-value(1 h, DAA) = 0.0034, $p$-value(6 h, C) = 0.0026, $p$-value(6 h, DAA) = 0.0029, +EGF: $p$-value(1 h) = 0.0028, $p$-value(6 h) = 0.0018, $n = 3$ biologically independent samples, by two-tailed Student's t-test). Scale bars represent 5 µm. Data are presented as mean values ± SEM. ns not significant; *$p < 0.05$, **$p < 0.01$, ***$p < 0.001$. Source data are provided as a Source Data file.

---

translation, until NSC differentiation is induced. However, in zebrafish, where the splicing sites of *scrt1a* and *scrt1b* introns meet the consensus, regulation at the level of splicing does not exist. Zebrafish *Scratch1* paralogs start to be expressed later in the neurogenic lineage, and the transcripts are directly exported to the cytoplasm, where they can be translated. These observations suggest that in mammals, intron detention might allow the generation of a reservoir of transcripts that accumulate in the nucleus until they are required. In this sense, the onset of *Scratch1* transcription when stem cells acquire the neural identity, would 'prime' these cells for differentiation, which will require the production of Scratch1 protein.

Consistently, we have shown that *Scratch1* mRNA is rapidly spliced and exported in response to the differentiation signal, being available for translation already 1 h after the cells received this extracellular input. Interestingly, ref. 51. showed that *Scratch1* is one of the first genes that are expressed when microglia is reprogrammed into neurons, suggesting that Scratch1 protein is required for the acquisition of neuronal fate[17–19,21]. In agreement with this, we have found that Scratch1 promotes NSC differentiation into neurons. Moreover, *Scratch1* downregulation also caused a significant decrease in the survival of the differentiating cells, indicating that this transcription factor protects cells from undergoing apoptosis, as shown for Scratch family members in *C. elegans*[18,22] and in zebrafish[23]. Remarkably, the reduction in the levels of *Scratch1* did not cause any phenotype in undifferentiated NSCs, in agreement with the nuclear accumulation of its transcripts, not being functional and, therefore, dispensable in these cells.

More importantly, here we show that the regulatory mechanism that controls *Scratch1* mRNA splicing and export also regulates the subcellular localisation of many other transcripts relevant for NSCs differentiation. Having a reservoir of mRNAs in the nucleus allows the rapid translation of these transcripts precisely when they are required, accelerating the response of the cells to external stimuli. In the case of immediate-early genes, it is known that their transcripts are shorter than average[52], reducing the time for translation. Conversely, intron detention has been associated with shorter introns[7,8] as also shown here, but with longer transcripts[7,13], providing a mechanism to generate rapid changes in their expression, despite the long length. Moreover, the finding that transcripts associated with maintaining NSCs in an undifferentiated state rapidly switch to intron detention in response to the differentiation signal, blocking their translation, dampens residual transcription and sharpens the response of the NSCs both in culture and in vivo.

Detained introns can be spliced in response to different stimuli[10–16], including the onset of differentiation in several contexts[12,15,16,53]. However, little is known about how extracellular signals are transduced to trigger the post- transcriptional processing of the transcripts. We show that, in the context of adult neurogenesis, the splicing of detained introns and the subsequent export to the

cytoplasm of the different subsets of transcripts is promoted by mRNA methylation, revealing that m⁶A modification acts as a connection between external stimuli and the processing of detained introns in response to them. In fact, most mRNA-modifying enzymes reside in the nucleus[34], where they have access to DI-mRNAs and their activity is known to be responsive to extracellular inputs[54]. Mechanistically, m⁶A can constitute a binding element for specific proteins and can also function as a switch modifying the secondary structure of the mRNA[43]. The latter occurs because the addition of the methyl group destabilises A-U pairs[55,56], hindering the formation of RNA duplexes[57] and exposing binding sites for RNA binding proteins, as has been demonstrated for factors involved in splicing[35,58]. In the case of *Scratch1* intron, we have found an enlargement of the PPT with sequence features that likely contributes to the regulation of intron detention; and that the processing of this intron is stimulated by RNA m⁶A modification. Considering this, it is possible that *Scratch1* mRNA methylation promotes its splicing by the destabilization of RNA duplexes involving the PPT, favouring the recognition of this intronic sequence by U2AF2.

Being dynamic states, intron detention and m⁶A RNA methylation can mark groups of functionally related transcripts to respond to changes in the extracellular environment in a synchronised manner, acting as an RNA regulon[59]. Our data indicate that m⁶A regulates the balance between NSC self-renewal and differentiation in the adult brain, and altogether, these findings emphasise the relevance of assessing post-transcriptional regulation in the control of numerous biological processes, in addition to transcriptome analysis. In the context of the adult NSC niche, our data provide a mechanism that solves the putative conflicting fate decisions due to the simultaneous transcription of stemness and differentiation genes, and that tightly controls the timing of adult neural differentiation.

## Methods
### Animal models
**Mice.** All experiments were performed using adult C57BL/6J (JAX™ Mice Strain) wild-type mice between 2- and 4-months-old as a source of biological samples. Mice were bread and housed in a temperature-controlled room under 12 h periods of light/darkness and were reared on standard chow and water *ad libitum*. All animal procedures were conducted in compliance with the European Community Council Directive (2010/63/EU) and Spanish legislation. The protocols were approved by the CSIC Ethical Committee and the Animal Welfare Committee at the Institute of Neurosciences (Alicante, Spain).

**Zebrafish.** All the experiments were performed using 6 months-old male zebrafish of the AB strain (European Zebrafish Resource Center, cat. no. 1175.1), which were maintained at 28 °C under standard conditions. All animal procedures were conducted in compliance with the

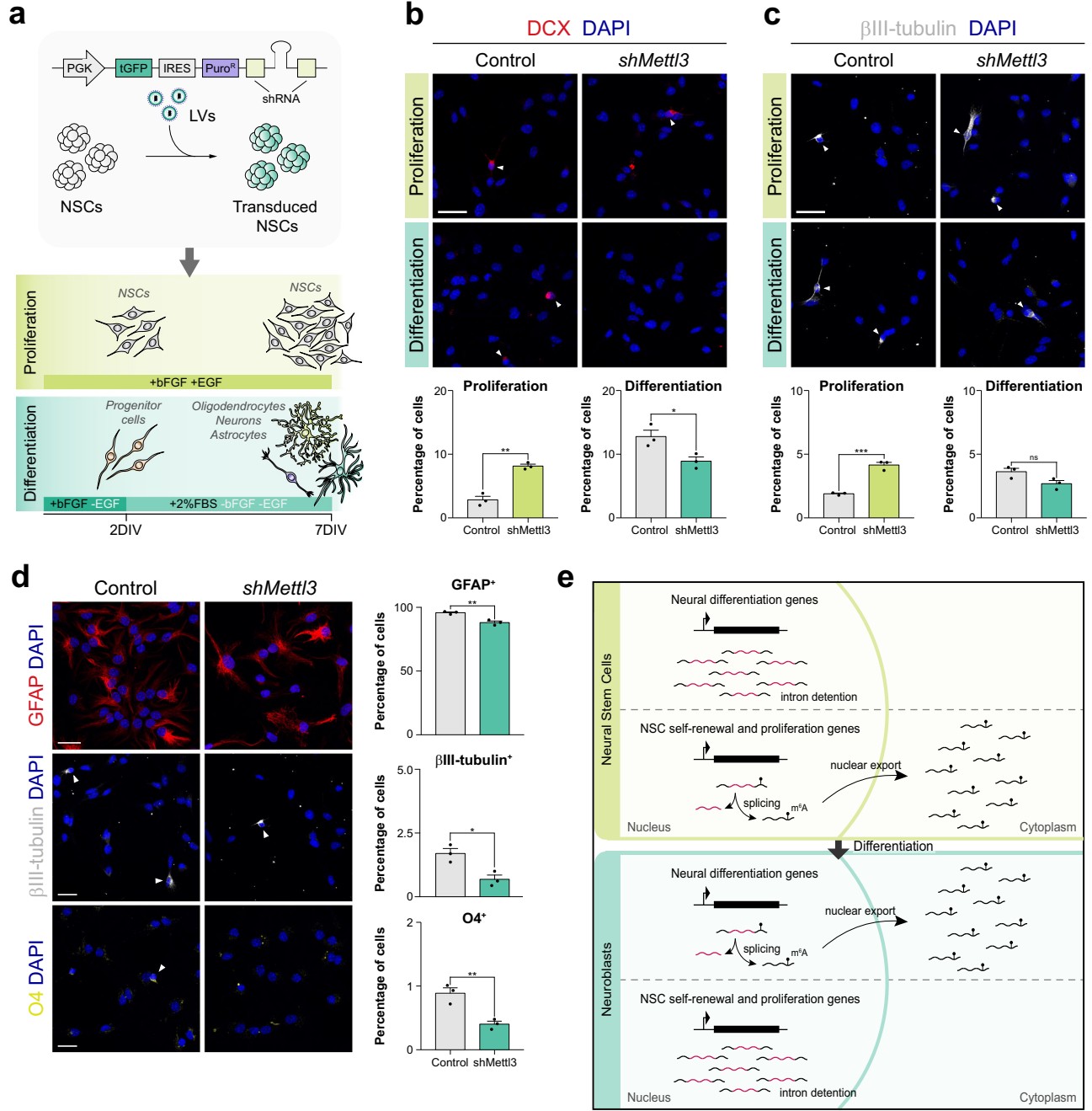

**Fig. 8 | RNA methylation regulates the balance between NSC self-renewal and differentiation. a** Schematic representation of the transduction of NSCs with lentiviruses (LV) and the construct used for *Mettl3* loss of function experiments. Transduced NSCs were subsequently cultured either under proliferation or differentiation conditions. **b** Immunodetection and quantification of DCX⁺ cells (red) in cultures previously infected with control or shMettl3 lentiviruses 2 days after plating the cells, both in proliferation and differentiation conditions (2DIV; *p*-value(proliferation) = 0.0011, *p*-value(differentiation) = 0.033, *n* = 3 biologically independent samples, by two-tailed Student's t-test). **c** Immunodetection and quantification of βIII-tubulin⁺ cells (white) in cultures previously infected with control or shMettl3 lentiviruses 2 days after plating the cells, both in proliferation and differentiation conditions (2DIV; *p*-value(proliferation) = 0.0005, *p*-value(differentiation) = 0.061, *n* = 3 biologically independent samples, by two-tailed

Student's t-test). **d** Immunodetection and quantification of GFAP⁺ (red; *p*-value = 0.005, *n* = 3 biologically independent samples, by two-tailed Student's t-test), βIII-tubulin⁺ (white; *p*-value = 0.017, *n* = 3 biologically independent samples, by two-tailed Student's t-test) and O4⁺ (yellow; *p*-value = 0.008, *n* = 3 biologically independent samples, by two-tailed Student's t-test) cells in differentiating cultures previously infected with control or shMettl3 lentiviruses 7 days after the induction of differentiation (7DIV). **e** Proposed model: Intron detention regulates the translational availability of multiple mRNAs associated with NSC self-renewal, proliferation and differentiation, contributing to the fine tuning of neural differentiation. Arrowheads point to positive cells. Scale bars represent 25 μm. Data are presented as mean values ± SEM. ns not significant; *$p < 0.05$, **$p < 0.01$, ***$p < 0.001$. Source data are provided as a Source Data file.

European Community Council Directive (2010/63/EU) and Spanish legislation. The protocols were approved by the CSIC Ethical Committee and the Animal Welfare Committee at the Institute of Neurosciences (Alicante).

## NSC culture and differentiation, drug treatment and lentiviral infection
Adult neurosphere cultures were generated as previously described in ref. 33. Briefly, 2- to 4-months-old animals were sacrificed by cervical

dislocation and both SEZs from each brain were dissected and enzymatically digested by incubation in Earle's balanced salt solution (EBSS; Gibco, cat. no. 24010-043) containing 12 U/ml papain (Sigma, cat. no. P3125), 0.2 mg/ml L-cystein (Sigma, cat. no. C8277) and 0.2 mg/ml EDTA (Sigma, cat. no. E6511) for 30 min at 37 °C. Tissue was then rinsed in Control media, composed of Dulbecco's modified Eagle's medium (DMEM; Sigma, cat. no. D6429)/F12 medium (1:1 v/v; Sigma, cat. no. N6658) containing 0.6% glucose (Sigma, cat. no. G7021), 0.1% NaHCO$_3$ (Sigma, cat. no. S5761), 5 mM HEPES (Sigma, cat. no. H3375), 2 mM L-glutamine (Gibco, cat. no. 25030-024), 100 U/ml Penicillin-Streptomycin (Gibco, cat. no. 15140-122), 80 µg/ml apo-transferrin (Sigma, cat. no. T2252), 50 nM insulin (Sigma, cat. no. I6634), 10 µg/ml putrescine (Sigma, cat. no. P7505), 0.02 nM progesterone (Sigma, cat. no. P6149), 30 nM sodium selenite (Sigma, cat. no. S9133) 0.7 U/ml heparin sodium salt (Sigma, cat. no. H3149) and 4 mg/ml Bovine Serum Albumin (BSA; Sigma, cat. no. B4287); and mechanically dissociated to a single-cell suspension. Isolated cells were collected by centrifugation, resuspended in Complete medium (Control medium supplemented with 20 ng/ml epidermal growth factor (EGF; Gibco, cat. no. 53003-018) and 10 ng/ml fibroblast growth factor (FGF; Sigma, cat. no. F0291) and incubated at 37 °C in a 5% CO$_2$ humidified incubator. For culture expansion, grown neurospheres were disaggregated by treatment with Accutase (Sigma, cat. no. A6964) for 10 min, followed by mechanical dissociation; and replated at a density of 10,000 cells/cm$^2$ in Complete medium.

For NSC differentiation, 40,000 individual cells/cm$^2$ were seeded in Matrigel-coated coverslips and incubated in Control medium supplemented only with 10 ng/ml FGF (without EGF) for not more than 2 days. For further differentiation, cells were incubated with fresh Control medium supplemented with 2% foetal bovine serum (FBS; Sigma, cat. no. F7524) for 5 more days.

When indicated, NSC cultures were treated with Gefitinib (Selleck Chemicals, S1025; 2.5 µM), Afatinib (Enzo Life Sciences, ENZ-CM158; 2.5 µM) or 3-Deazaadenosine (DAA, Sigma, D8296; 100 µM). In the cases when NSCs were fixed before 12 h after treatment or induction of differentiation, they were first seeded in normal Complete medium for 24 h and then treated as required.

Lentivirus containing *Scratch1*-specific shRNAs (RMM4532-EG170729 (Clones V3LMM_ 456558 and V3LMM_ 456561), Open Biosystems), a *Mettl3*-specific shRNA (RMM4532-EG56335, Open Biosystems) or a non-silencing control (RHS4346, Open Biosystems) pGIPZ plasmids were generated in HEK293T packing cells (ATCC, cat. no. ACS-4500) and concentrated by ultracentrifugation in an Optima XL-100K centrifuge (Beckman Coulter), as previously described in ref. 60. For overexpression studies, mouse *Scratch1* cDNA was cloned in the pRRL-SIN-PPT.PGK.EGFP.WPRE expression vector, followed by P2A and tdTomato fluorescent protein. For NSC infection, grown neurospheres were disaggregated and incubated with $5 \times 10^6$ TU/ml lentiviral particles for 6 h. NSC were then rinsed in Control medium and incubated in fresh Complete medium at 37 °C in a 5% CO$_2$ humidified incubator, allowing them to grow for following experiments.

## ESC differentiation

The mouse embryonic stem cell (ESC) line 46 C (Sox1-GFP-IRES-pac knock-in; PrimCells, cat. no. PCEMM01) was cultured on 0.1% gelatin-coated plates in ESC medium (Glasgow Minimum Essential Medium (GMEM; Sigma, cat. no. G5154)) supplemented with 10% FBS (ES tested; Capricorn, cat. no. FBS-12A), 2mM L-glutamine (Gibco, cat. no. 25030-024), sodium pyruvate (Gibco, cat. no. 11360-070), 0.1 mM non-essential amino acids (NEAA; Gibco, cat. no. 1140-050), 100 U/ml Penicillin-Streptomycin (Gibco, cat. no. 15140-122), 0.1 mM β-mercaptoethanol (Sigma, cat. no. M3148) and 10 ng/ml leukaemia inhibitory factor (LIF; Prepotech, cat. no. AF-300-05). Cells were incubated at 37 °C in a 5% CO$_2$ humidified incubator.

ES cells were differentiated as previously described in ref. 61. Briefly, 20,000 cells/cm$^2$ were seeded on gelatin-coated plates in N2B27 medium (DMEM (Sigma, cat. no. D6429)/F12 (Sigma, cat. no. N6658) medium (1:1 v/v) supplemented with modified N2 (25 µg/ml bovine insulin (Sigma, cat. no. I6634), 100 µg/ml apo-transferrin (Sigma, cat. no. T2252), 16 µg/ml putrescine (Sigma, cat. no. P7505), 6 ng/ml progesterone (Sigma, cat. no. P6149), 30 nM sodium selenite (Sigma, cat. no. S9133), 100 U/ml Penicillin-Streptomycin (Gibco, cat. no. 15140-122) and 50 µg/ml bovine serum albumin (fraction V; Gibco, cat. no. 15260-037), combined 1:1 with Neurobasal medium (Life technologies, cat. no. 21103-049) supplemented with B27 (Gibco, cat. no. 17504-044) for 7 days, refreshing the medium each day. From day 6 on, cells were treated with 0.5 µg/ml puromycin (Calbiochem, cat. no. 540411) to select the cells that had undergone neural lineage specification (Sox1$^+$ cells). Cells were re-plated 48 h after the addition of puromycin into uncoated culture dishes in NS medium (EuroMed-N medium (Euroclone, cat. no. AN-18-113) supplemented with 2 mM L-glutamine (Gibco, cat. no. 25030-024), N2 (freshly prepared as previously described), 10 ng/ml EGF (Gibco, cat. no. 5003-018) and 10 ng/ml FGF (Sigma, cat. no. F0291) in the absence of puromycin for 2–3 days, until neurosphere-like aggregates were formed. Then, aggregates were collected and dissociated and individual cells were plated on gelatin-coated plates with fresh NS medium.

Induced pluripotent stem cells (iPSC) reprogrammed from NSC were kindly provided by Sacri R. Ferrón.

## Immunocytochemistry

Cells were fixed with 2% paraformaldehyde (PFA) for 15 min and incubated in blocking solution (10% FBS, 1% glycine and 0.1% Triton X-100, only when required, in 0.1 M Phosphate-buffered Saline pH 7.4 (PBS)) for 45 min and then with the indicated primary antibodies (see Supplementary Table 1) overnight (o/n) a 4 °C. After several washes with PBS, cells were incubated with the appropriate fluorescently labelled secondary antibodies (see Supplementary Table 1) for 45 min at room temperature (RT), counterstained with DAPI (Sigma, cat. no. D9542; 1 µg/ml) and mounted with Mowiol. Samples were imaged with a Leica DMR microscope (Leica) and analysed with ImageJ software. Confocal images were obtained in a FV1200 confocal microscope (Olympus).

## In situ hybridisation

In all cases, in situ hybridisation (ISH) was performed using digoxigenin (DIG)-labelled probes (see Supplementary Data 1) synthesised by in vitro transcription of the indicated DNA templates, as previously described in ref. 62.

In order to obtain adult mouse brain sections, 2- to 4-months-old animals were anaesthetised and transcardially perfused with diethyl pyrocarbonate (DEPC; Sigma, cat. no. D5758)-treated PBS, followed by 4% PFA in PBS-DEPC, at a flow rate of 5.5 ml/min. Brains were then extracted, post-fixed o/n in the same fixative and serially sectioned into 50 µm coronal slices using a Leica VT1000 vibratome (Leica). Sections were dehydrated through a series of increasing methanol concentration (25%, 50%, 75% and 100%) in PBS-DEPC 0.1% Tween 20 (PBT), rehydrated through methanol:PBT in reverse order and finally washed with PBT. Next, sections were incubated in 1% hydrogen peroxide (H$_2$O$_2$) for 40 min, washed with PBT, treated with 10 µg/ml proteinase K for 3 min at RT, re-fixed with 4%PFA and washed again with PBT. Sections were then incubated with prehybridisation solution (50% formamide (Sigma, cat. no. 47671), 5x SSC, 2% Roche blocking powder (Roche, cat. no. RD1096176), 0.1% Tween20 (Sigma, cat no. P9416), 50 mg/ml heparin (Sigma, cat. no. H5263), 1 mg/ml tRNA (Roche, cat. no. RD109495), 1 mM EDTA (Sigma, cat. no. E5134), 0.1% CHAPS (Sigma, cat. no. C3023) in DEPC-treated dH$_2$O) at 60 °C for 1 h first and then o/n after refreshing prehybridisation solution. Sections were either used the next day for ISH or stored at −20 °C.

Then, prehybridized sections were incubated with 1 µg/ml of denatured DIG- labelled probes o/n at 60 °C. Next day they were washed twice with 2x SSC, 0.1% CHAPS, then twice with 0.2x SSC, 0.1% CHAPS and finally with KTBT washing buffer (50 mM Tris- HCl pH7.5, 150 mM NaCl, 10 mM KCl, 0.1% Triton X-100 (Sigma, cat. no. T8787) in H₂O). After the washes, sections were incubated in blocking solution (20% goat serum, 0.7% Roche blocking solution in KTBT) for 3 h at 4 °C. For probe detection, sections were then incubated with 1:500 dilution of an anti-DIG peroxidase (POD)-conjugated antibody and with the indicated antibodies for immunohistochemistry (IHC; see Supplementary Table 1) in blocking solution o/n at 4 °C. The whole next day embryos were washed with KTBT several times. For developing, sections were incubated with Amplification solution (TSA® fluorescein detection kit; PerkinElmer, cat. no. NEL744001KT) for 1 min, following manufacturer's instructions. Fluorescein isothiocyanate (FITC; 1:100) was added to the Amplification solution and sections were incubated in the mix for 45 min in dark at RT. After washing the sections with KTBT, they were incubated with the appropriate fluorescently labelled secondary antibodies (see Supplementary Table 1) for 1 h at RT. Finally, sections were washed, counterstained with DAPI and imaged with Olympus FV1200 confocal microscope (Olympus).

To perform in situ hybridisation on cell cultures, cells were seeded in Matrigel-coated coverslips and fixed when indicated with prewarmed 2% PFA-DEPC for 15 min. After several washes with PBT, samples were incubated in 1% H₂O₂ for 20 min at RT, washed again with PBT and incubated with prehybridisation solution at 60 °C for 1 h first and then o/n after refreshing prehybridisation solution. Samples were either used the next day for ISH or stored at −20 °C. Next, cells were incubated with 1 µg/ml of denatured DIG-labelled probes o/n at 60 °C and, next day, washed three times with the first washing solution (50% formamide, 5x SSC, 1% SDS (Sigma, cat. no. L4509) in H₂O) at 60 °C, then three times with the second washing solution (50% formamide, 2x SSC in H₂O) at 60 °C and finally three times with TBST (140 mM NaCl, 2.7 mM KCl, 25 mM Tris-HCl pH7.5, 0.1% Tween20 in H₂O) at RT. For fluorescent in situ hybridisation, cells were then incubated with blocking solution (10% FBS in TBST) for 1 h at RT and after with 1:500 dilution of an anti-DIG POD-conjugated antibody as well as with the indicated antibodies for immunocytochemistry (ICC; see Supplementary Table 1) in blocking solution o/n at 4 °C. After a whole day of TBST washes, cells were incubated with Amplification Solution (Perkin Elmer, cat. no. NEL744001KT) for 1 min to adjust the pH and then with FITC (1:100) diluted in Amplification Solution for 15 min in dark at RT. Then, cells were washed several times with TBST, incubated with the corresponding secondary antibodies (see Supplementary Table 1), washed again with TBST, counterstained with DAPI and imaged with an Olympus FV1200 confocal microscope (Olympus). For visible in situ hybridisation, cells were incubated with blocking solution (10% FBS in TBST) for 1 h at RT and then incubated with 1:1000 alkaline phosphatase (AP)-conjugated anti-DIG antibody (see Supplementary Table 1) diluted in blocking solution o/n at 4 °C. After a whole day of TBST washes, cells were washed several times with NTMT (100 mM Tris-HCl pH9.5, 59 mM MgCl₂, 100 mM NaCl, 0.1% Tween-20, 1 mM levamisole in H₂O) and incubated with NTMT containing freshly added 3 µl NBT and 2.6 µl BCIP per 1 ml (developing solution), in the dark at RT until the colour reaction develops. After obtaining the desired signal level, cells were washed several times with TBST and fixed with 4% PFA. Samples were imaged using a Leica DMR microscope.

For DNA in situ hybridisation, cells were plated on 0.2% gelatin-coated glass coverslips and, after they attached to the matrix, cells were fixed with 2% PFA for 15 min at RT. Next, they were washed three times with PBS, incubated with 0.2% pepsin for 4 min at 37 °C and post-fixed with 4% PFA for 5 min at RT. After washing twice with PBS, cells were dehydrated by incubating them with 70%, 90% and 100% ethanol, 3 min each; and after the coverslips were air dried for several minutes. In the meantime, chromosome 15 specific probe labelled with biotin

(Chrombios®, cat. no. PM15BI) was denatured by incubating it for 6 min at 72 °C. Then, cells were incubated with the denatured probe o/n at 60 °C. The day after, samples were washed once with 2x SSC at 60 °C, three times with 50% formamide 2x SSC at 60 °C and twice with Tris-Saline-Tween buffer (TST; 1x saline, 0.1 M Tris, 0.05% Tween20 in H₂O) at RT. After that, cells were incubated with blocking solution (TSBSA; 1x saline, 0.1 M Tris, 20% BSA in H₂O) for 1 h at RT and then with 1:500 dilution of Alexa Fluor™ 568-conjugated Streptavidin (see Supplementary Table 1). Finally, samples were washed several times with TST, stained with DAPI and imaged with an Olympus FV1200 confocal microscope.

## Total RNA extraction, cDNA synthesis and qPCR analyses

For gene expression assays, total RNA was extracted using the Illustra RNAspin Mini isolation kit (GE healthcare, cat. no. 25-0500-70), following manufacturer's instructions. Reverse transcription was carried out using the Maxima First Strand cDNA Synthesis kit (Thermo Scientific, cat. no. 10334500). Quantitative RT-PCR was performed in a Step One Plus machine (Applied Biosystems) using Fast SYBR Green Mastermix (Applied Biosystems, cat. no. 10556555) and the primers listed in Supplementary Data 2. Relative levels of expression were calculated using the comparative Ct method normalised to the internal control *TBP* housekeeping transcript.

## RNA immunoprecipitation

Total RNA was extracted using mirVana miRNA Isolation Kit (Invitrogen, cat. no. 10763147). RNA immunoprecipitation (RIP) was carried out as described in ref. 63, with minor modifications. Briefly, 60 µl of protein-A magnetic beads (BioRad, cat. no. 161-4013) were blocked in 0.5 mg/ml BSA (Sigma, cat. no. A8022) plus 2 µg/ml salmon sperm DNA (Applied Biosystems, cat. no. AM9680) solution for 2 h at 4 °C and incubated with 10 µg/ml anti-N⁶-methyladenosine (m⁶A) antibody (Sigma, cat. no. ABE572) or IgG control antibody (Diagenode, cat. no. C15410206; see Supplementary Table 1) o/n at 4 °C with head-over-tail rotation. After several washes in IP buffer (150 mM NaCl, 10 mM Tris-HCl, pH 7.5, 0.1% IGEPAL CA-630 (Sigma, cat. no. I8896) in nuclease free H₂O), antibody-bound beads were incubated with 2 µg of purified RNA for 4 h at 4 °C, in the presence of 10 µl/ml RNasin ribonuclease inhibitor (Promega, cat. no. N2111). Then, beads were washed twice in IP buffer, twice in low-salt IP buffer (50 mM NaCl, 10 mM Tris-HCl, pH 7.5, 0.1% IGEPAL CA-630 in nuclease free H₂O) and twice in high-salt IP buffer (500 mM NaCl, 10 mM Tris-HCl, pH 7.5, 0.1% IGEPAL CA-630 in nuclease free H₂O) for 10 min each at 4 °C. m⁶A-enriched RNA was then eluted in TRIzol Reagent (Invitrogen, cat. no. 15596-018) and isolated following manufacturer's instructions.

Reverse transcription and quantitative RT-PCR were performed as described above (see Supplementary Data 2 for the list of primers used). The expression percentage of a target gene in IP sample was calculated relative to that in input sample and normalised to the relative expression of *GAPDH* transcript.

## Detection of potential m⁶A sites

m⁶A site prediction was performed form primary *Mus musculus* *Scratch1* RNA sequence (ENSMUST00000096365.4, Ensembl) using TargetM6A method (http://csbio.njust.edu.cn/bioinf/TargetM6A)[64].

## RNA-seq, splicing isoforms analysis and functional enrichment analyses

Library preparation and high-throughput sequencing were performed by the Genomic Unit at the Centre for Genomic Regulation (CRG). Paired-end read (125 bp length) libraries for assayed conditions were produced using HiSeq v4 Chemistry kit (Illustra, cat no. FC-401-4003) and processed with the sequencer software HiSeq Control Software version 2.2.58. Sequencing quality was checked using FastQC (Babraham Institute). Read alignment and gene count were performed using

STAR version 2.5.3[65] against *Mus musculus* genome assembly mm10 (GRCm38 build; Ensembl). Differential gene expression analysis was performed with DESeq2 (v. 1.22.1)[66] using a Wald Test and *p*-values were adjusted for multiple testing using Benjamini-Hochberg FDR correction. RNA-seq data were deposited at the Gene Expression Omnibus (GEO) under the accession number GSE180806.

Functional enrichment analysis: We used the enrichR R package (v.2.1) to access the Enrichr database[67] and performed general gene functional enrichment analysis. ReactomePA R package (v.1.30.0)[68] was used to perform Reactome pathway gene set enrichment analysis, while the gseGO and gseKEGG functions in clusterProfiler R package (v.3.10.0)[69] were used to carry out Gene Set Enrichment Analysis (GSEA) of Gene Ontology terms[70] and Kyoto Encyclopaedia of Genes and Genomes[71] pathways.

Alternative splicing genome wide quantification was performed using VAST-TOOLS (v.2.0.2)[72] against *M. musculus* mm10 (GRCm38 build; Ensembl). Complementary analysis of intron retention was performed with ASpli R/Bioconductor package (v.2.0.0)[73]. After selecting the 2000 IR events with the highest mean absolute difference among the three time-points, IR scores were scaled by performing a z-score normalisation on every value. Cluster analysis was performed using hierarchical clustering with the heatmap.2 function in the R package gplots (v.3.0.1.1) (https://cran.r-project.org/web/packages/gplots/index.html), and soft clustering using the fuzzy c-means algorithm in Mfuzz R package (v. 2.42.0)[74] after selecting the 2000 IR events with the highest mean absolute difference among the three time-points.

Mouse genome annotation was obtained from GENCODE (release M25, GRCm38), and intron coordinates were extracted using the gtf2leafcutter.pl script from LeafCutter[75]. Intron fasta sequences were obtained with getfasta from bedtools (v.2.27.1)[76] and their GC content was calculated with nuc from bedtools (v.2.27.1)[76]. ggpubr R package v.0.4.0 (https://CRAN.R-project.org/package=ggpubr) was used to generate intron plots and statistics (two-tailed t-test).

### Quantification and statistical analysis

All statistical analyses were carried out using Prism (GraphPad software, version 8.0.2). For cell counting and quantitative PCR experiments, treatments were compared to their corresponding controls using unpaired two-tailed Student's t-test. All bar graphs represent mean ± SEM (Standard Error of the Mean). Statistical significances were as follows: $* = P \leq 0.05$, $** = P \leq 0.01$ and $*** = P \leq 0.001$.

### Reporting summary

Further information on research design is available in the Nature Portfolio Reporting Summary linked to this article.

## Data availability

The data supporting the findings of this study are available from the corresponding authors upon request. The RNA-seq data generated in this study have been deposited in the GEO database under accession code GSE180806. The RNA-seq data generated in ref. 4 are available in the GEO databased under the accession code GSE94991. Source data are provided with this paper.

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

## Acknowledgements

We thank Cristina López-Blau for mouse care and technical help, Diana Abad for helpful technical input, Sonia Vega for her help and support in managing cell lines, Giovanna Expósito and Verona Villar Cerviño for the support at the imaging facility at IN, and the members of M.A.N. laboratory for continuous and helpful discussions. We thank Sacri R. Ferrón for kindly providing iPSCs samples and CRG Core Technologies Programme for their support and assistance in the splicing optimised sequencing experiments. This work was supported by grants to M.A.N. from the Spanish Ministry of Science and Innovation (MCI PID2021-125682NB-I00), Instituto de Salud Carlos III (CIBERER/19/3.1), and Generalitat Valenciana (GVA) (ISIC 2012/010 plus Prometeo Programme 2017/150 and 2021/045). Work in I.F.'s lab was supported by MICIU grants PID2020-117937GB-I00, RED2018- 102723-T, CB06/05/0086 (CIBERNED), and RD16/0011/0017 (RETIC Tercel) plus Prometeo 2017/030. Work in J.V.'s lab was supported by MICIU PID2020-114630GB-I00 and ERC-AdG-LS2-670146I and the Generalitat de Catalunya through the CERCA programme. M.A.N. and J.V. also acknowledge financial support from the Spanish State Research Agency, through the 'Severo Ochoa Programme' for Centres of Excellence in R&D, Grants CEX2021-001165-S and CEX2020-001049-S, respectively. A.G.-I. was holder of a FPU predoctoral contract from the Spanish Ministry of Education, Culture and Sports.

## Author contributions

A.G.-I. performed most experiments, analysed and interpreted the data, prepared the figures and wrote the manuscript. A.D.-M. helped to perform some experiments in I.F.'s laboratory. A.A. performed bioinformatics analyses. E.M. performed bioinformatics analyses in J.V.'s laboratory. J.G. examined the sequences of Scratch genes in different species and contributed to interpret the data. J.V. designed RNA-seq experiments and contributed to interpret the data. I.F. contributed to design some experiments and interpret the data. M.A.N. conceived the project, interpreted the data, wrote the manuscript and secured funding.

## Competing interests

The authors declare no competing interests.
