## [Peer Review File · Nature Communications]

Intron detention tightly regulates the stemness/differentiation switch in the adult neurogenic nicheREVIEWER COMMENTS

Reviewer #1 (Remarks to the Author):

In this manuscript, Nieto's group proposes intron detention tightly regulates the stemness/differentiation switch in the adult neurogenic niche. Using FISH, the authors show *scratch1*, as an exemplary differentiating gene, is retained in the nucleus of NSCs, and transported to the cytoplasm upon differentiation. The authors propose this is caused by intron detention in NSCs, indicated by FISH using the intron probe, but the causality is not established. The authors further show that differentiation increases m6A of *scratch1*, which the authors propose promotes splicing. Through RNA-seq, the authors identify additional genes increasing intron splicing, and another gene group decreasing intron splicing upon differentiation. Therefore, the authors suggest this is a widespread regulatory mechanism controlling stemness/differentiation.

Overall, the paper presents an interesting post-transcriptional regulation concerning intron detention during adult neurogenesis. The splicing changes are evident by FISH and RNA-Seq (though the magnitude is less clear, which casts a shadow on their biological impacts). However, the characterization is mostly descriptive. The proposed mechanistic link, m6A-splicing-nuclear export-protein expression, is supported only by correlative data. Furthermore, the impact of this regulation on neuronal differentiation is not demonstrated, so the authors should change the title and tone down their conclusions in the text.

Major:

The data show the splicing changes of selective genes during neuronal differentiation, but it remains unclear to what degree introns are detained to prevent expression. In situ is great but do not discern the ratio between spliced and unspliced transcripts. The intron probe may stain only a small proportion of total nuclear transcripts. The authors should use RT-PCR to quantify the degree of intron detention, e.g., during differentiation for *scratch1*, and various candidate genes, to give readers an idea of the level of splicing changes.

Related to the above question, there is an alternative explanation to the authors' data: the major mechanism of preventing protein expression of differentiated genes is splicing-independent nuclear retention of spliced transcripts. In other words, to support the authors' conclusion (on intron detention), can the authors directly stimulate *scratch1* intron splicing in NSCs to induce its protein expression (and subsequent differentiation)? Or some other kind of experiments to indicate causality (e.g., manipulating the polypyrimidine tract as the authors proposed)?

Fig 6: do the authors observe the same intron detention events in published datasets of neuronal differentiation (particularly in vivo datasets, e.g., GSE94991)? Or how well is this phenomenon reproduced? For the same genes? this is an important test.

Fig6, how are the IR scores scaled? What are the range of splicing changes without scaling?

Fig3. The authors' model would predict splicing and cytoplasmic transport do not occur in glia cells after differentiation. Is that true?

Fig4: Does genetic inhibition of m6A modification (e.g., via manipulating *mettl3/14*) affect *Scratch1* splicing and nuclear export? This is important because DAA is not necessarily very specific.

Important to connect the dots in this paper, can the authors provide insight into how genes are selectively modified by m6A? and how m6A modification changes during differentiation?

The FISH images are great, but how representative are they? Does every cell look as shown? The authors should quantify percentages of cells.

Minor:

An important assumption of the paper is that *Scratch1* protein expression would impair stem cell maintenance. Has this been shown? For example, by cDNA transfection? Does promoting *Scratch1* splicing impair stem cell maintenance?

experimental replicates are not indicated in any fig legends

The authors should disclose the probe sequences for the in situ hybridization experiments.

The idea of intron detention switch at the onset of differentiation has been documented before (e.g., PMID: 30496473). Can the authors draw similarity and/or differences in discussion?

Reviewer #2 (Remarks to the Author):

In this manuscript the authors describe intron retention as a level of fast regulation of transcripts in NSC to neuroblast transition. They first describe this for Scratch1 that is retained in the nucleus, but fast spliced and released from the nucleus in differentiation upon EGF withdrawal. This process is correlated to increase m6A and blocking RNA methylation by DAA block release of the transcript from the nucleus. Most importantly, the authors show by RNAseq that there are other transcripts behaving similar to Scratch1 highlighting that splicing regulation precedes transcriptome changes by several hours upon EGF withdrawal. This work thus unravelled a novel mechanism of regulating NSC differentiation, namely by intron retention and nuclear retention.

- 1) My main concern with this manuscript is the lack of quantification and statistics. All the data on retention of transcripts in the nucleus are based on single examples. The authors need to show quantification of cells with transcripts retracted to the nucleus indicating also the number of biological replicates and an adequate statistical test. Without this, the observations remain anecdotal.
 - 2) To verify if cells have cytoplasmic extensions at all into which the transcripts could spread, the authors need to provide phase contrast pictures or a simultaneous cytoplasmic marker, at least for their core results.
 - 3) For the few data presented in histograms please show each data point as a dot, and mention the number of independent biological replicates in the Figure legend.
- Once data can be trusted on the basis of statistical analysis, this is a beautiful manuscript providing key new insights into NSC regulation.

Reviewer #3 (Remarks to the Author):

González-Iglesias et al. reported intron retained pre-mRNAs of Scratch1 and other differentiation genes in NSC located in nucleus and might involve in NSC stemness and differentiation. However, the experiments are not well designed, and several key questions are not demonstrated.

- (1) The authors found Scratch1 mRNA accumulates in the nucleus of NSCs due to intron retention. However, whether and how the intron retention of Scratch1 mRNA plays a role in NSC stemness maintenance has not been fully addressed. It is also the same for other intron-retained mRNAs, such as Chd5, Ptpv.
- (2) The m6A modification has been proposed as a potential regulation mechanism of intron retention. The experiment of DAA treatment suggested a potential role of m6A on splicing regulation, however, direct evidence, such as mutation assays, minigene reporter, is lacking. What's more, how m6A prevents intron retention of Scratch mRNA is unknown.
- (3) Why does m6A on Scratch1 pre-mRNA elevate during neural differentiation? What's the underlying mechanism? How does it contribute to NSC differentiation?
- (4) The authors detected intron-retained Scratch1 mRNA by in situ hybridization all through the manuscript. Although intron-specific probe was used, the authors should provide enough positive and negative control, and at least use another method to examine the isoform switch of Scratch mRNA.

As a general statement, we first wanted to apologise for the unusual long time that this revision has taken. When we submitted and received the comments from the referees we were still in pandemic times and although we still cannot fully understand the reason, our cultures started to fail to work for over one full year. It was in part due to the difficulties in receiving products from the companies we used to use before the pandemic and we also believe that there were additional intrinsic problems in our facility, something that we had not faced before. As we wanted to address all the referees' comments, we have had to wait until we could robustly and reliably culture the NSCs as previously.

See below a point-by-point response to comments and suggestions. All revisions in the manuscript are highlighted in blue to facilitate identification.

Reviewer #1

Overall, the paper presents an interesting post-transcriptional regulation concerning intron detention during adult neurogenesis. The splicing changes are evident by FISH and RNA-Seq (though the magnitude is less clear, which casts a shadow on their biological impacts). However, the characterization is mostly descriptive. The proposed mechanistic link, m6A-splicing-nuclear export-protein expression, is supported only by correlative data. Furthermore, the impact of this regulation on neuronal differentiation is not demonstrated, so the authors should change the title and tone down their conclusions in the text.

We thank the reviewer for finding our data interesting and for suggesting experiments that have allowed us to provide further evidence to confirm the mechanism proposed. As stated above, we want to apologise for the long time that has taken us to submit this revised version. All revisions in the manuscript are highlighted in blue to facilitate identification.

Major:

- The data show the splicing changes of selective genes during neuronal differentiation, but it remains unclear to what degree introns are detained to prevent expression. In situ is great but do not discern the ratio between spliced and unspliced transcripts. The intron probe may stain only a small proportion of total nuclear transcripts. The authors should use RT-PCR to quantify the degree of intron

detention, e.g., during differentiation for *scratch1*, and various candidate genes, to give readers an idea of the level of splicing changes.

Thanks very much for the suggestion. In response, we assessed the ratio of spliced mRNA compared to total by RT-qPCR in NSCs during differentiation (Figs. 4d, e, and 7i, j). We are pleased to report that our conclusion was correct and there was a massive retention of *Scratch1* mRNA in NSCs. Upon receiving the differentiation signal, transcripts are spliced and exported to the cytoplasm. We have also quantified it after *Scratch1* cDNA overexpression (Figs. 5l and S4f) and upon the inhibition of m⁶A deposition (Figs. 5f and S5e), the latter in response to other comments (see below) and the results are consistent with the mechanism proposed. We also show that detained introns is the mechanism of regulation for other candidate genes (see Figs. S8-S10).

- Related to the above question, there is an alternative explanation to the authors' data: the major mechanism of preventing protein expression of differentiated genes is splicing-independent nuclear retention of spliced transcripts. In other words, to support the authors conclusion (on intron detention), can the authors directly stimulate *scratch1* intron splicing in NSCs to induce its protein expression (and subsequent differentiation)? Or some other kind of experiments to indicate causality (e.g., manipulating the polypyrimidine tract as the authors proposed)?

Thanks. This suggestion has allowed us to confirm that it is intron detention what causes *Scratch1* mRNA nuclear accumulation. We have overexpressed *Scratch1* cDNA in NSC primary cultures and found that, in addition to the transcripts retained in the nucleus, also detectable in control conditions, other transcripts exhibited a cytoplasmic distribution, as expected from mRNAs expressed from *Scratch1* cDNA that do not contain intronic sequences and can be immediately exported (Fig. 1g-i). This indicates that intron retention prevents the nuclear export of *Scratch1* mRNA in NSCs. The same was observed when *Scratch1* overexpressing cultures were treated with DAA during differentiation, preventing m⁶A deposition (Fig. 5j, l).

- Fig 6: do the authors observe the same intron detention events in published datasets of neuronal differentiation (particularly in vivo datasets, e.g., GSE94991)? Or how well is this phenomenon reproduced? For the same genes? this is an important test.

Thanks again for the suggestion. We have analysed the suggested RNA-seq datasets obtained from NSCs, early neuroblasts (ENB) and late neuroblasts (LNB)

isolated from the SEZ or OB of adult mice in a beautiful work by Ana Martin-Villalba's lab (Ref 4; Baser et al., 2019). We found thousands of IR events. Although in many cases these IR events did not involve the same genes, probably due to the differences in the time-points analysed, similar splicing patterns were detected and, importantly, functional enrichment analysis showed that equivalent clusters contained genes associated with the same processes (Fig. S7).

- Fig6, how are the IR scores scaled? What are the range of splicing changes without scaling?

After selecting the 2000 IR events with the highest mean absolute difference among the three time-points, IR scores are scaled by performing a z-score normalization on every value. The plots depicted in Fig. 6b and 6e show not scaled values.

- Fig3. The authors' model would predict splicing and cytoplasmic transport do not occur in glia cells after differentiation. Is that true?

Thanks for the question. We apologise if we were not clear. The regulatory mechanism described in this study operates during neural differentiation, affecting both neuronal and glial cells. As such, we show that intron detention controls the expression of *Sox6*, a gene involved in astrocyte and oligodendrocyte differentiation (Figs. 7a, b and S8a).

- Fig4: Does genetic inhibition of m⁶A modification (e.g., via manipulating *mettl3/14*) affect *Scratch1* splicing and nuclear export? This is important because DAA is not necessarily very specific.

We again thank the referee for this question, that has allowed us to confirm that m⁶A modification was responsible for the regulation of *Scratch1* mRNA splicing. As happened when cultures were treated with DAA, the inhibition of m⁶A deposition by *Mettl3* downregulation prevented the splicing and nuclear export of *Scratch1* transcripts (Figure S5) and also of other mRNAs belonging to Cluster 1 (Fig. S10a). In the case of Cluster 2 mRNAs, *Mettl3* downregulation induced intron retention and nuclear accumulation of the transcripts in NSCs (Fig. S10b), as it happened in response to DAA treatment.

- Important to connect the dots in this paper, can the authors provide insight into how genes are selectively modified by m⁶A? and how m⁶A modification changes during differentiation?

Little is known about the mechanisms that control m⁶A specificity, and further research will be required to understand the underlying mechanism that controls the switch in m⁶A deposition during NSC differentiation. The recent publication by Luo et al. (Nat Comm **13**, 2720, 2022) illustrates well the complexity of sequence determinants of m⁶A deposition, well beyond -in distance and complexity- RRACH motifs, as revealed by deep learning models. We respectfully believe that this lies out of the scope of this manuscript.

- The FISH images are great, but how representative are they? Does every cell look as shown? The authors should quantify percentages of cells.

The images already included in the previous version were fully representative of the process observed. The quantifications provide now further confirmation that this is the case. We quantified mRNA distribution and the splicing status in our *in situ* hybridisation experiments. Samples were obtained in each experiment from 3 different NSC primary cultures, and we examined at least 100 cells per condition and biological replicate. To facilitate the interpretation of the images and quantification of mRNA distribution, we now include in Fig. 1d the XZ orthogonal projection of confocal images. This projection shows that analysis can perfectly identify the subcellular localization of transcripts and faithfully discriminate between nucleus or cytoplasm. We thank the reviewer for the suggestion as, although it meant a significant effort, it provides further compelling evidence of the transcripts' nuclear retention.

Minor:

- An important assumption of the paper is that Scratch1 protein expression would impair stem cell maintenance. Has this been shown? For example, by cDNA transfection? Does promoting Scratch1 splicing impair stem cell maintenance?

Yes, the referee is right. Upon *Scratch1* cDNA overexpression we observed an increase in the proportion of neurons generated when NSCs differentiation was induced (7DIV, Fig. 2k). Moreover, *Scratch1* cDNA overexpression induced an increase in the percentage of DCX⁺ and β III-tubulin⁺ cells 48 hours after the induction of differentiation (Figs. S3c, d, bottom panels) and also when cultures were kept in

proliferation conditions (Figs. S3c, d, top panels), indicating that, as suggested, Scratch1 impairs stem cell maintenance in NSC.

- Experimental replicates are not indicated in any fig legends

We apologise for this overlook. Figure legends were revised to include the number of replicates.

- The authors should disclose the probe sequences for the *in situ* hybridization experiments.

The corresponding sequences of *in situ* hybridization probes are now shown in Supplementary Table 2.

- The idea of intron detention switch at the onset of differentiation has been documented before (e.g., PMID: 30496473). Can the authors draw similarity and/or differences in discussion?

It has been shown before that detained introns are spliced at the onset of differentiation in several contexts^{12, 15, 16, 53}, although it had not been described before during neural differentiation. Additionally, this study provides new insights in the regulation of detained introns processing, connecting for the first time their splicing with m⁶A deposition and also showing the reciprocal regulation between differentiation and stemness genes.

Reviewer #2

In this manuscript the authors describe intron detention as a level of fast regulation of transcripts in NSC to neuroblast transition. They first describe this for Scratch1 that is retained in the nucleus, but fast spliced and released from the nucleus in differentiation upon EGF withdrawal. This process is correlated to increase m⁶A and blocking RNA methylation by DAA block release of the transcript from the nucleus. Most importantly, the authors show by RNAseq that there are other transcripts behaving similar to Scratch1 highlighting that splicing regulation precedes transcriptome changes by several hours upon EGF withdrawal. This work thus unravelled a novel mechanism of regulating NSC differentiation, namely by intron detention and nuclear retention.

We thank the reviewer for stating that our work unravels a novel mechanism in a process as relevant as the regulation of NSC differentiation. On the other hand, as mentioned in our response to reviewer 1, we want to apologise for the long time that has taken us to submit this revised version.

1) My main concern with this manuscript is the lack of quantification and statistics. All the data on retention of transcripts in the nucleus are based on single examples. The authors need to show quantification of cells with transcripts restricted to the nucleus indicating also the number of biological replicates and an adequate statistical test. Without this, the observations remain anecdotal.

We cannot thank the reviewer enough for requesting a deep and detailed quantification analysis. We have now quantified mRNA distribution and splicing status in all experiments all throughout the manuscript. To facilitate the interpretation of the images and quantification of mRNA distribution, we now include in Fig. 1d the XZ orthogonal projection of confocal images. This projection shows that analysis can perfectly identify the subcellular localization of transcripts and faithfully discriminate between nucleus or cytoplasm. Our *in situ* hybridisation samples were obtained in each experiment (all figures throughout the manuscript) from 3 different NSC primary cultures, and we examined at least 100 cells per condition and biological replicate. This has involved a very significant amount of work, but the data are clear-cut and allow to fully confirm our previous conclusions.

2) To verify if cells have cytoplasmic extensions at all into which the transcripts could spread, the authors need to provide phase contrast pictures or a simultaneous cytoplasmic marker, at least for their core results.

To better visualise the cellular distribution of transcripts, we have carried out a series of *Scratch1* and *Scratch2* visible *in situ* hybridisations and provide bright field images shown in Figs. S1d, e.

3) For the few data presented in histograms please show each data point as a dot, and mention the number of independent biological replicates in the Figure legend. Once data can be trusted on the basis of statistical analysis, this is a beautiful manuscript providing key new insights into NSC regulation.

Thanks for the suggestion and sorry for the overlook. We should have included this information in the previous version. The number of biological replicates is now

indicated in the figure legends and each data point is represented as a dot in the histograms associated with immunofluorescence and *in situ* hybridisation quantifications. We are also very grateful to the reviewer for the nice words about our study.

Reviewer #3

González-Iglesias et al. reported intron retained pre-mRNAs of *Scratch1* and other differentiation genes in NSC located in nucleus and might involve in NSC stemness and differentiation. However, the experiments are not well designed, and several key questions are not demonstrated.

We were sorry to hear that the reviewer found some experiments not well designed, and as specific examples were not stated, we understand that this led to the statement that some important issues were not demonstrated. We now address the reviewer's comments and suggestions and show functional analyses that confirm our previous proposal. On the other hand, as mentioned at the beginning of this letter, we want to apologise for the long time that has taken us to submit this revised version due to unexpected and still not well understood (but fully solved) technical problems in our cell culture facility during and after pandemic that we had never faced before. All revisions in the manuscript are highlighted in blue to facilitate identification.

(1) The authors found *Scratch1* mRNA accumulates in the nucleus of NSCs due to intron retention. However, whether and how the intron retention of *Scratch1* mRNA plays a role in NSC stemness maintenance has not been fully addressed. It is also the same for other intron-retained mRNAs, such as *Chd5*, *Ptpv*.

We have addressed this general concern by performing a series of different functional experiments.

- Firstly, we overexpressed *Scratch1* cDNA and we observed an increase in the proportion of neurons generated when NSCs differentiation was induced (7DIV, Fig. 2k). Moreover, *Scratch1* cDNA overexpression induced an increase in the percentage of DCX⁺ and β III-tubulin⁺ cells 48 hours after the induction of differentiation (Figs. S3c, d, bottom panels) and also when

cultures were kept in proliferation conditions (Figs. S3c, d, top panels), indicating that *Scratch1* function impairs stemness maintenance in NSC.

- We also examined the effect of inhibiting m⁶A deposition by *Mettl3* downregulation in NSC primary cultures. In agreement with the proposed model, we found that *Mettl3* downregulation disrupted the balance between NSC self-renewal and differentiation, as reflected by the decrease of DCX+ and β III-tubulin+ cells in the cultures upon the induction of differentiation (Fig. 8b, c, bottom panels) and their increase when cells were kept in proliferation conditions (Fig. 8b, c, top panels).

(2) The m⁶A modification has been proposed as a potential regulation mechanism of intron retention. The experiment of DAA treatment suggested a potential role of m⁶A on splicing regulation, however, direct evidence, such as mutation assays, minigene reporter, is lacking. What's more, how m⁶A prevents intron retention of *Scratch* mRNA is unknown.

Thanks for the suggestion. We have now overexpressed *Scratch1* cDNA in NSC primary cultures and found that, in addition to the transcripts retained in the nucleus, also detectable in control conditions, other transcripts exhibited a cytoplasmic distribution, as expected from mRNAs expressed from *Scratch1* cDNA that do not contain intronic sequences and can be immediately exported. The same was observed during differentiation when *Scratch1* overexpressing cultures were treated with DAA, preventing m⁶A deposition (Figs. 5j, l). These data indicate that it is intron retention what causes *Scratch1* mRNA nuclear accumulation.

Mechanistically, it has been previously shown that m⁶A can constitute a binding element for specific proteins and can also function as a switch modifying the secondary structure of the mRNA⁴³. The latter is due to the fact that the addition of the methyl group destabilizes A-U pairs^{55,56}, which hinders the formation of RNA duplexes⁵⁷ and exposes binding sites for RNA binding proteins, as has been demonstrated for factors involved in splicing^{35,58}. In the case of *Scratch1* intron, we have found an enlargement of the PPT with particular sequence features that likely contributes to the regulation of intron retention; and that the processing of this intron is stimulated by RNA m⁶A modification. Considering this, it is possible that *Scratch1* mRNA methylation promotes its splicing by the destabilization of RNA duplexes involving the PPT, favoring the recognition of this intronic sequence by U2AF2. Interestingly, this effect of m⁶A deposition is novel for splice site recognition and

distinct from the repression of U2AF1 binding upon methylation of the 3' splice site AG reported by Mendel et al (Cell **184**, 3125, 2021).

Finally, we also confirm m⁶A deposition as the mechanism to promote splicing and mRNA nuclear export. This is demonstrated by knocking down *Mettl3*, which prevented both m⁶A deposition and splicing, and nuclear export of *Scratch1* transcripts (Figure S5) and of other mRNAs belonging to Cluster 1 and Cluster 2 (Fig. S10).

(3) Why does m⁶A on *Scratch1* pre-mRNA elevate during neural differentiation? What's the underlying mechanism? How does it contribute to NSC differentiation?

We now provide further data on the contribution of m⁶A deposition in the regulation of NSC differentiation. As such, *Mettl3* downregulation disrupted the balance between NSC self-renewal and differentiation. In addition, the inhibition of m⁶A deposition also caused a reduction in the generation of astrocytes, neurons and oligodendrocytes compared to control cultures (Fig. 8d). Little is known about the mechanisms that control m⁶A specificity, and further research will be required to understand the underlying mechanism that controls the switch in m⁶A deposition during NSC differentiation. The recent publication by Luo et al (Nat Comm **13**, 2720, 2022) illustrates well the complexity of sequence determinants of m⁶A deposition, well beyond -in distance and complexity- RRACH motifs, as revealed by deep learning models. We respectfully believe that this lies out of the scope of this manuscript.

(4) The authors detected intron-retent *Scratch1* mRNA by in site hybridization all through the manuscript. Although intron-specific probe was used, the authors should provide enough positive and negative control, and at least use another method to examine the isoform switch of *Scratch* mRNA.

We thank the reviewer for proposing the use of an additional method to evaluate the splicing status of the transcripts analysed. We have assessed the ratio of spliced mRNA compared to total by RT-qPCR in NSCs during differentiation (Figs. 4d, e, and 7i, j), after *Scratch1* cDNA was overexpressed (Figs. 1k, 5l and S4f) and upon the inhibition of m⁶A deposition (Figs. 5f and S5e). The results are consistent with the mechanism proposed. We also show that detained introns is the mechanism of regulation for other candidate genes (see Figs. S8-S10).

REVIEWERS' COMMENTS

Reviewer #1 (Remarks to the Author):

The authors have provided new experiments and data to address my previous concerns. One suggestion about the presentation of the new analysis comparing the authors' datasets and those of Baser et al. 2019. It is valuable, as the authors stated in the manuscript, that IR regulations identified by both datasets shared broadly-defined GO terms. It is equally valuable for readers to know that the IR-regulated genes between the two datasets are largely not overlapping even for similarly-trended clusters (based on the current presentation) and that Scatch1 is not listed in the Baser et al. results. The incomplete overlap is understandable but needs to be pointed out. The Fig6e mentioned in the rebuttal letter is missing in the manuscript.

Reviewer #2 (Remarks to the Author):

I am happy to see that the authors managed to reproduce the cultures of NSCs and improve the manuscript by showing biological replicates and convincing Z-projections and examples. Unfortunately the individual data points are still missing in Supplemental Figures 1a,f, 4f (grey bars), 5b, e (grey bars), 12b as well as in the main Figure 3b,e,f,g, 4d,e, 5d,e,i, l, 7i,j. I also suggest to mention the statistical test used in the Figure legends. For the Mettl3 downregulation by the shRNA it would be nice to show protein in addition to RNA levels. There is also a typo in the heading of the panels in Supplemental Figures 1d and 2 (scractch instead of Scratch). With these corrections and additions the manuscript is ready for publication.

RESPONSE TO REVIEWERS' COMMENTS

Reviewer #1 (Remarks to the Author):

The authors have provided new experiments and data to address my previous concerns.

*One suggestion about the presentation of the new analysis comparing the authors' datasets and those of Baser et al. 2019. It is valuable, as the authors stated in the manuscript, that IR regulations identified by both datasets shared broadly-defined GO terms. It is equally valuable for readers to know that the IR-regulated genes between the two datasets are largely not overlapping even for similarly-trended clusters (based on the current presentation) and that *Scatch1* is not listed in the Baser et al. results. The incomplete overlap is understandable but needs to be pointed out.*

We thank the reviewer for the suggestion. This has been included in the new version of the manuscript (page 12).

The Fig6e mentioned in the rebuttal letter is missing in the manuscript.

We apologize for the overlook. We meant new Figs. 7b and 7d. Corrected.

Reviewer #2 (Remarks to the Author):

I am happy to see that the authors managed to reproduce the cultures of NSCs and improve the manuscript by showing biological replicates and convincing Z-projections and examples.

Unfortunately the individual data points are still missing in Supplemental Figures 1a,f, 4f (grey bars), 5b, e (grey bars), 12b as well as in the main Figure 3b,e,f,g, 4d,e, 5d,e,i, l, 7i,j.

We apologize for the missing info. We have now included the individual data points in all the graphs.

I also suggest to mention the statistical test used in the Figure legends.

Done. A mention to the statistical test used has been included in the Figure legends when appropriate.

For the Mettl3 downregulation by the shRNA it would be nice to show protein in addition to RNA levels.

We agree with the reviewer that having protein levels would be a nice addition. This experiment would mean performing a whole set of new experiments that would take a significant amount of time, even if the antibodies would work optimally from the first attempt. We also understand that this was a suggestion rather than a requirement and, since there is no specific mention to this experiment on your letter, we have not performed it.

There is also a typo in the heading of the panels in Supplemental Figures 1d and 2 (scractch instead of Scratch).

We thank the reviewer for spotting this. We have corrected both typos.

With these corrections and additions the manuscript is ready for publication.